# A ferroptosis-targeting ceria anchored halloysite as orally drug delivery system for radiation colitis therapy

Yue Feng[1,7], Xiang Luo[2,3,4,7], Zichun Li[2,3,4], Xinjuan Fan[5,6], Yiting Wang[5,6], Rong-Rong He [2,3,4,8] ✉ & Mingxian Liu [1,8] ✉

Radiation colitis is the leading cause of diarrhea and hematochezia in pelvic radiotherapy patients. This work advances the pathogenesis of radiation colitis from the perspective of ferroptosis. An oral Pickering emulsion is stabilized with halloysite clay nanotubes to alleviate radiation colitis by inhibiting ferroptosis. Ceria nanozyme grown in situ on nanotubes can scavenge reactive oxygen species, and deferiprone was loaded into the lumen of nanotubes to relieve iron stress. These two strategies effectively inhibit lipid peroxidation and rescue ferroptosis in the intestinal microenvironment. The clay nanotubes play a critical role as either a medicine to alleviate colitis, a nanocarrier that targets the inflamed colon by electrostatic adsorption, or an interfacial stabilizer for emulsions. This ferroptosis-based strategy was effective in vitro and in vivo, providing a prospective candidate for radiotherapy protection via rational regulation of specific oxidative stress.

As a radiation-sensitive organ, the intestine is susceptible to injury during pelvic radiotherapy, leading to radiation colitis with diarrhea and hematochezia[1]. The prevalence of inflammatory bowel disease has surpassed that, seriously endangering patients' lives. The mechanism is that radiation-induced hydrolysis generates reactive oxygen species (ROS), such as superoxide anion radicals ($\cdot O_2^-$), hydroxyl radicals ($\cdot OH$), and $H_2O_2$, which trigger a variety of cell death-related pathways. The intestinal cell death damages the mucosal barrier and activates further inflammation and bleeding[2]. Although the mechanisms of irradiation damage (especially apoptosis) are well known, clinical radioprotection measures are often ineffective, and new treatment modalities are worth investigating. The efficacy of apoptosis-inhibiting agents such as 5-androstenediol and the free radical trapping agent amphotericin is unsatisfactory. Radioprotective agents alone also fail

to alleviate patient complications such as inflammation, diarrhea, and intestinal bleeding in specific diseases such as radiation colitis. Therefore, it is vital to elucidate the mechanisms of the progression of radiation colitis and design a drug delivery strategy for precise treatment with simultaneous symptom relief.

Various radiation-induced tissue injuries are firmly related to ferroptosis[3]. Ferroptosis during irradiation mainly involves a regulated cell death induced by iron-dependent lipid peroxidation and massive accumulation of ROS[4,5]. Subsequently, the immune pathways are activated, and the recruited macrophages scavenge the dead cells after recognizing the ferroptosis cell membrane through their surface receptors[6]. Furthermore, bleeding results in the phagocytosis of massive erythrocytes by macrophages, which induces iron release from heme and its accumulation in the cells, also inducing ferroptosis[7].

[1]Department of Materials Science and Engineering, College of Chemistry and Materials Science, Jinan University, 511443 Guangzhou, China. [2]Guangdong Engineering Research Center of Chinese Medicine & Disease Susceptibility, Jinan University, 510632 Guangzhou, China. [3]International Cooperative Laboratory of Traditional Chinese Medicine Modernization and Innovative Drug Development of Chinese Ministry of Education (MOE), College of Pharmacy, Jinan University, 510632 Guangzhou, China. [4]Guangdong Province Key Laboratory of Pharmacodynamic Constituents of TCM and New Drugs Research, College of Pharmacy, Jinan University, 510632 Guangzhou, China. [5]Department of Pathology, The Sixth Affiliated Hospital, Sun Yat-sen University, 510655 Guangzhou, China. [6]Guangdong Provincial Key Laboratory of Colorectal and Pelvic Floor Diseases, The Sixth Affiliated Hospital, Sun Yat-sen University, 510655 Guangzhou, China. [7]These authors contributed equally: Yue Feng, Xiang Luo. [8]These authors jointly supervised this work: Rong-Rong He, Mingxian Liu. ✉e-mail: rongronghe@jnu.edu.cn; liumx@jnu.edu.cn

The oxidative stress is further exacerbated by the Fenton reaction of iron encountering high levels of $H_2O_2$ at the site of inflammation[8]. The evidence mentioned above implies that ferroptosis may have a crucial role in radiation colitis, so scavenging ROS, inhibiting bleeding, and relieving overloaded iron through drugs are expected to mitigate radiation colitis.

Nanozymes with ROS scavenging properties are gaining attention due to their catalytic efficiency, multi-enzyme activity, and gastrointestinal stability. Among them, ceria nanoparticles ($CeO_2$) are multi-enzyme mimics with superoxide dismutase (SOD) and catalase (CAT) like activities as well as hydroxyl radical scavenging activity. These catalytic antioxidant activities enable $CeO_2$ to alleviate ROS-induced diseases such as radiation injury, inflammation, neurodegenerative diseases, etc. For example, $CeO_2$/zeolite imidazole ester skeleton composite material can treat oxidative injury induced by reperfusion in ischemic stroke[9]. Increasing oxygen vacancies by modulating the surface strain of $CeO_2/Mn_3O_4$ nanocrystals can scavenge irradiation-induced ROS injury to protect intestinal stem cells[10]. Oral administration of $CeO_2$/montmorillonite hybrid can achieve antioxidant synergistic antidiarrheal treatment of colitis[11].

Mucosal defects in the intestine are often accompanied by local inflammation due to infection by intestinal contents, causing the accumulation of positively charged proteins such as transferrin and antimicrobial peptide, which allows negatively charged materials to concentrate in the inflamed area of the mucosal defect through electrostatic interactions[12]. Halloysite clay nanotubes (HNTs), a traditional mineral in Chinese medicine, have been used to stop bleeding and treat diarrhea for thousands of years[13,14]. Similar to montmorillonite clay (commercial name of Smecta®), HNTs are extremely promising for the treatment of colitis. It is a hollow tubular nanoclay formed by the mismatch of silica-oxygen tetrahedra and aluminum-oxygen octahedra. The positively charged $AlO_2(OH)_4$ layer is rolled inside, and the negatively charged $SiO_2$ layer is exposed on the outer surfaces. Overall, HNTs are highly negatively charged, which gives HNTs a candidate for targeting the inflamed intestine.

Moreover, the surface property of HNTs satisfies the energy and siting requirements for retaining metal ions. This retention mainly arises from electrostatic interactions, chemical bonding in surface complexation, and precipitation, facilitating surface redox reactions and precipitation of various metal ions[15]. At the same time, the hollow tubular structure and high aspect ratio render HNTs an excellent vehicle for drug protection, long residence time, and precise drug delivery[16]. In a word, HNTs were considered able to control diarrhea and hematochezia, and their tubular exterior is especially suitable for nanozyme growth for ROS scavenging. In addition, their tubular interior is an ideal drug-loading lumen, and their strong negative charge properties can mediate inflammatory site targeting.

Herein, ferroptosis was demonstrated to play a critical role in radiation colitis, and scavenging ROS and reducing iron stress are expected to alleviate radiation colitis (Fig. 1). By in situ growing $CeO_2$ outside the nanotubes and loading ferroptosis inhibitor deferiprone (DFP) in the lumen, a long-term stable Pickering emulsion with vitamin E (VE) was prepared ($CeO_2$@HNTs@DFP@VE, named CHDV). Furthermore, an oral nanoplatform was developed to alleviate radiation colitis effectively. HNTs serve the leading role since they act as medicinal clay for hemostasis and antidiarrhea, nanocarrier for inflammation targeting, and interfacial stabilizers for Pickering emulsions. This

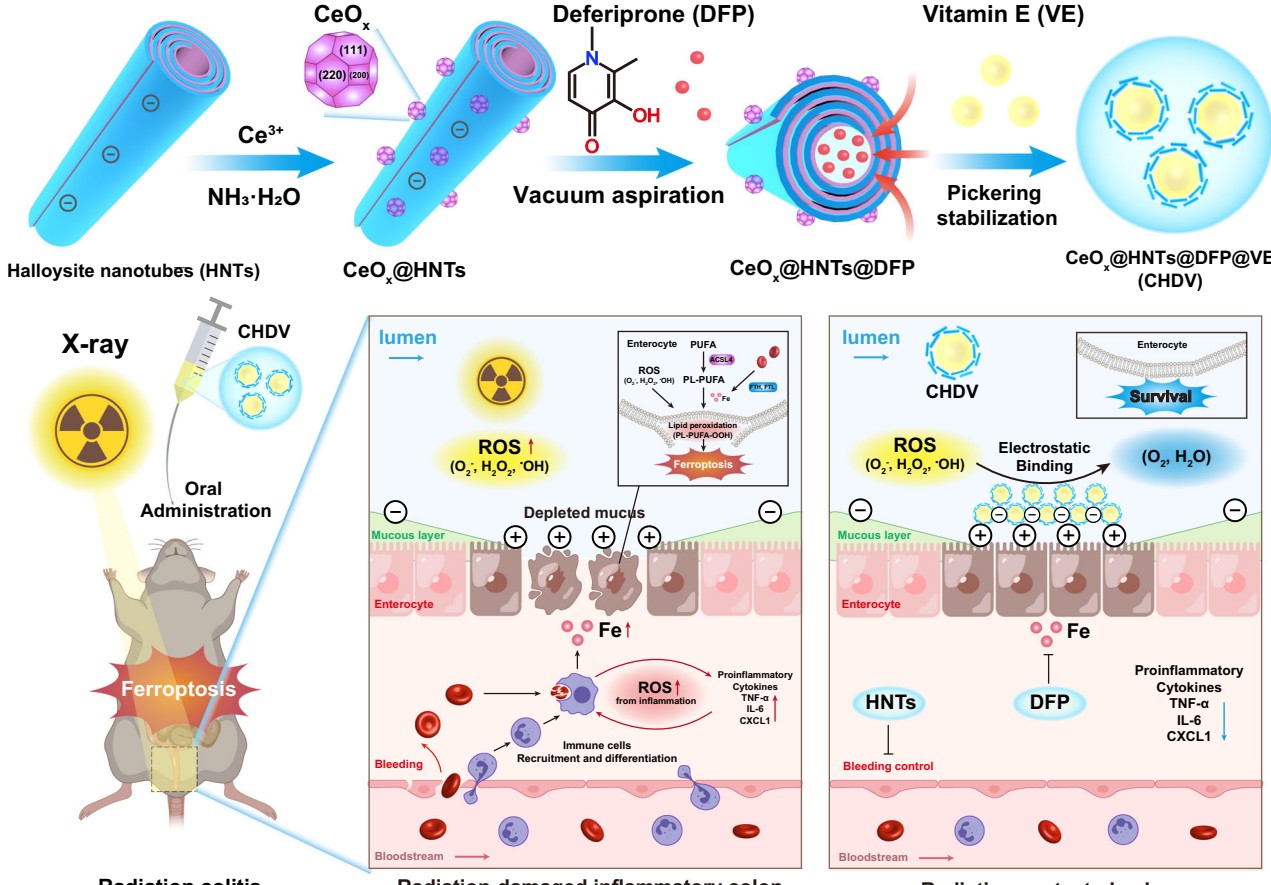

**Fig. 1 | Schematic illustration of the synthetic method and therapeutic mechanisms of CHDV Pickering emulsion.** As an oral nanoplatform, CHDV Pickering emulsion can treat radiation colitis via scavenging ROS and relieving ferroptosis.

oral nanoplatform achieves multiple synergistic ferroptosis inhibition, thereby effectively relieving radiation colitis.

## Results and discussion

### Ferroptosis mechanism of radiation colitis

Death of terminally differentiated enterocytes is a pathogenic factor in the development of intestinal injury[17]. It is well known that one of the significant irradiation-induced intracellular damages is the initiation of apoptosis due to ROS-induced DNA double-strand breaks[18]. However, anti-apoptotic therapy alone is usually of limited effectiveness. Accumulating evidence suggests that increased iron and lipid peroxidation may play crucial roles in the pathogenesis of irradiated intestine injury[19,20]. In this work, it was found that irradiation directly led to lipid peroxidation by inducing ROS production. Overexpression of Acyl-CoA synthase long-chain family member 4 (ACSL4) under oxidative stress mediates iron-dependent peroxidation of polyunsaturated fatty acid (PUFA)-containing phospholipids (PUFA-PLs) in the plasma membrane. This series of events initiated the cellular ferroptosis pathway. Intestinal bleeding exacerbated the iron overload of the inflammatory microenvironment with the participation of macrophages (Fig. 2a).

To investigate the correlation between ferroptosis and radiation colitis, we analyzed the protein expression of the ferroptosis process in colonic sections from a retrospective cohort of patients who underwent radiation colitis. The immunohistochemical analysis revealed a significant upregulation of 4-Hydroxynonenal (4-HNE), an end product of ferroptosis, as well as ACSL4. Representative results from patient with the most severe radiation colitis symptoms are presented in Fig. 2b. Data from the remaining patients are shown in Supplementary Fig. 1, and the complete clinical features are provided in Supplementary Table 3. These results indicate that classic ferroptosis biomarkers increase in correlation with the severity of the lesion sites, which indicates ferroptosis is an etiologic factor in radiation colitis.

To further study the alterations in the ferroptosis signaling pathway, protein expression in the mice model of radiation colitis were observed. It was found that ACSL4, ferritin heavy chain (FTH), ferritin light chain (FTL), and 4-HNE expression were significantly upregulated with increasing irradiation dose. (Fig. 2c, d). Quantitative real-time PCR (q-PCR) results of prostaglandin-endoperoxide synthase 2 (*Ptgs2*), *Acsl4*, *Ftl*, and ferritin heavy chain 1 (*Fth1*) also confirmed these trends (Fig. 2e), suggesting that ferroptosis may have a potentially critical role in irradiation-induced intestinal injury and show a positive correlation with irradiation dose. It remains unclear how irradiation induces upregulation of ACSL4 and may involve transcription factors or chromatin-modifying enzymes that regulate radiation response and ferroptosis[21]. Moreover, ACSL4 upregulation was found to be gut-specific but not in liver or lung tissue, and antimicrobial agents could revert irradiation-induced ASCL4 expression, speculating that infection and inflammation may be associated with irradiation-induced ACSL4 upregulation[5]. Ferritins (FTH and FTL) are responsible for storing and detoxifying cytosolic iron, and their upregulation confirms the active radiation-induced iron metabolism[22]. Furthermore, 4-HNE is a lipid peroxidation product that can trigger cytotoxicity and cell death in many cell types[23]. This series of events were found to be severe with increasing irradiation dose (Fig. 2c–e), suggesting an inextricable link between radiotherapy injury and ferroptosis.

Intestinal epithelial cells are highly radiosensitive, and their death results in the absence of an intestinal tight junction barrier, which is a leading cause of radiation colitis[2]. Therefore, irradiation-induced cell death modes were investigated and rescued to alleviate radiation colitis. When intestinal epithelial cells were exposed to irradiation, an immediate increase in cellular ROS was observed (Supplementary Fig. 2a, b). Previous studies often focused on apoptosis (irradiation-induced irreversible DNA double-strand breaks that trigger

programmed cell death)[10]. The accrue of γ-H2AX (a sensitive molecular marker of DNA damage and repair) after irradiation was compared (Supplementary Fig. 2c). Irradiation-induced lipid peroxidation can be significantly reduced by the ferroptosis inhibitor ferrostatin-1 (Fer-1), thereby partially rescuing the radiotherapy damage (Supplementary Fig. 3). Although ferroptosis plays a critical role, especially in irradiation-resistant cancer cells that are insensitive to DNA damage, the types of irradiation-induced cell death are diverse and include apoptosis, autophagy, and others[24].

Previous reports showed that irradiation-induced toxicity is primarily mediated through p53/puma-mediated apoptosis. So, a comparative study of ferroptosis vs apoptosis was performed. Immunofluorescence images of irradiated colon sections show apoptosis occurring over a large area of the intestinal epithelium, more in the villi than in the crypts. Ferroptosis also occurred extensively, but more intensely at the severely damaged mucosa and the tip of the colonic villi exposed to chyme (Supplementary Fig. 4a). Subsequently, reversal of radiation injury by inhibition of ferroptosis/apoptosis was assessed by Fer-1 and apoptosis inhibitor Z-VAD-FMK (Z-VAD), respectively (Supplementary Fig. 4b, c). It was found that they both partially reversed the cell death and plasma membrane damage caused by irradiation[25]. This phenomenon suggests that apoptosis and ferroptosis may coexist in irradiated cells, showing different death pathways due to the heterogeneity of cell tolerance. Since irradiation induces apoptosis is well established, this work focuses on the ferroptosis mechanism for radiation colitis treatment.

Ferroptosis occurs strongly at sites where the villi and tight junctions are severely damaged (Fig. 2f), suggesting that immunity is highly involved. Immunofluorescence co-staining of macrophages (red) and lipid peroxidation product 4-HNE (green) in the radiation colitis (7 days post irradiation exposure) tissue was performed. Significant 4-HNE accumulation and macrophage recruitment was observed in the colitis lesions. The accumulation and site variability of 4-HNE indicated that ferroptosis occurred at the tissue injury sites, and its abundance was positively correlated with the severity of the injury of villi and tight junctions. These severely injured sites activated inflammation leading to macrophage recruitment and activation. Notably, co-localization of macrophages and 4-HNE fluorescence was rarely observed, suggesting that ferroptosis did not occur in these recruited macrophages. Combined with the ferritin gene protein changes in the irradiated colon (Fig. 2b–e), ferroptosis may correlate with capillary rupture bleeding at the site of severe injury. These erythrocyte phagocytoses by macrophages causes hemoglobin-induced iron deposition and exacerbates oxidative stress and lipid peroxidation[7]. Moreover, the lipid peroxidation products from ferroptotic cells trigger an eat-me signal to recruit macrophages to clear dying cells[6].

By examining the anatomical regions occurring ferroptosis in the immunofluorescence images (Supplementary Fig. 4a, right), the terminally differentiated enterocytes, which are highly absorptive, are the main cell type undergoing ferroptosis. They are most abundant and are the main sites of ion uptake[26]. The uptake and translocation of iron lead to high iron loading and greater susceptibility to ferroptosis in response to irradiation-induced oxidative stress. Therefore, the radiation responsiveness of enterocytes in the cell model was further studied. Given the dramatic upregulation of ferroptosis-related biomarkers in the colon tissue of irradiated mice, the ferroptosis pathway may play a key role in induced enterocyte death. Irradiation-induced ROS were lethal, which brought decreased cell viability (Fig. 2g) and increased LDH release (an indicator of cell membrane damage). LDH release was observed to be stronger than the decrease in cell viability, implying that breakage of the cell membrane is a preemptive cause of irradiation-induced death (Fig. 2h). It indicates that death related to membrane damage may be the dominant factor of irradiation-induced death, not just apoptosis. In the apoptotic process, till the cell is

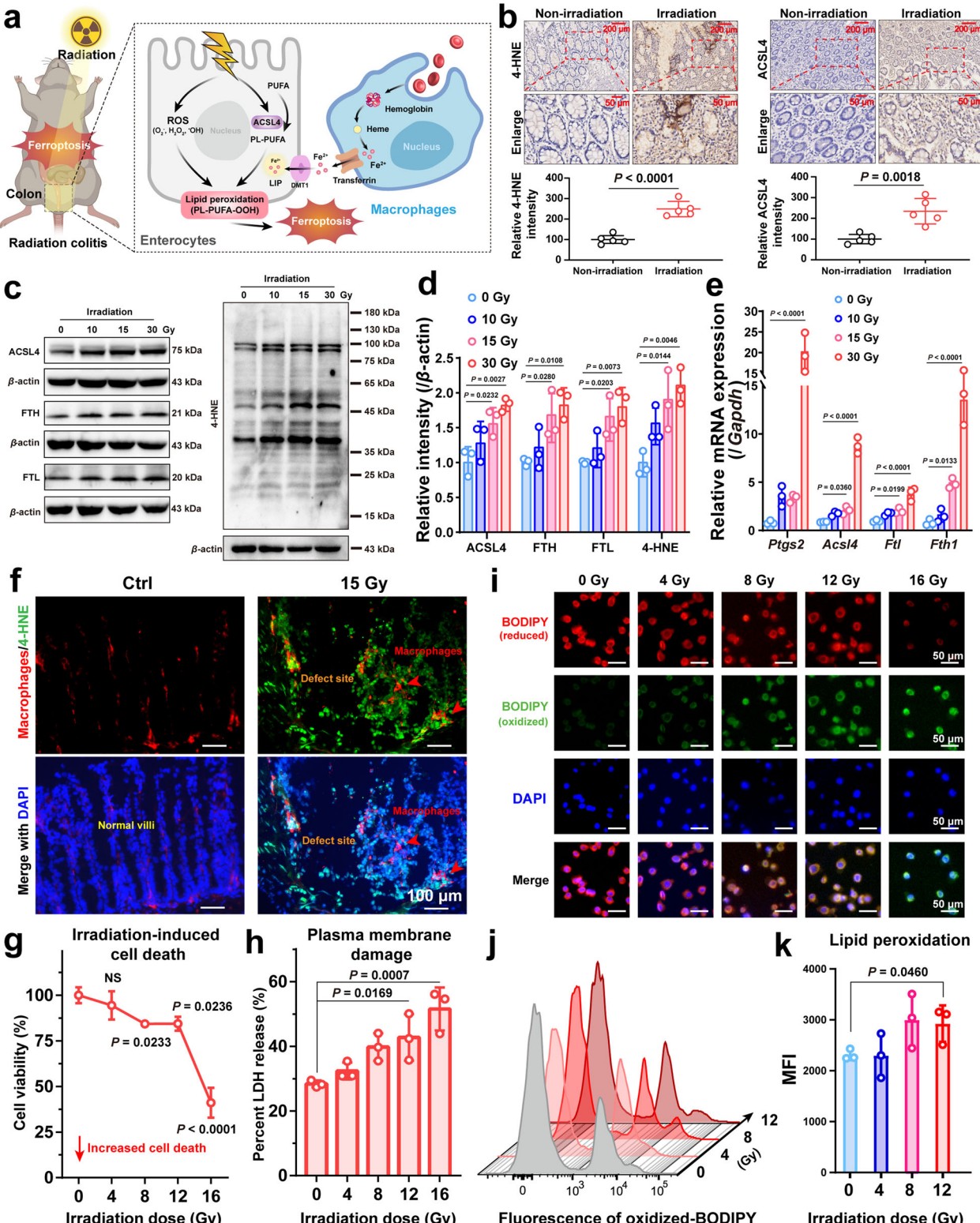

**Fig. 2 | Ferroptosis occurs in radiation colitis. a** Schematic showing the ferroptosis-related pathways under radiation colitis. **b** Representative immunohistochemical staining of 4-HNE and ACSL4 of colonic tissue samples from patients with radiation colitis, and their normalized expression intensity (relative to non-irradiation), $n = 5$. Scale = 200 μm (top panels) and 50 μm (bottom panels). $P$ values were calculated using a two-tailed unpaired Students $t$ test. **c** Western blotting analysis of ferroptosis-related protein expression in colon tissue of mice. **d** Normalized protein expression (relative to β-actin) levels, $n = 3$. **e** The ferroptosis-related gene expressions of *Ptgs2, Acsl4, Ftl,* and *Fth1* in colon tissue, $n = 3$. **f** Immunofluorescence images (F4/80, 4-HNE, DAPI) of colon tissue. Scale

bars, 100 μm. **g** Cell viability ($n = 3$), and **h** lactate dehydrogenase (LDH) release after incremental dose irradiation exposure ($n = 3$). **i** Fluorescence images of reduced (red) and oxidized (green) C11-BODIPY (scale bars, 50 μm), **j** representative flow cytometry histogram of oxidized C11-BODIPY (10,000 cells per tube were collected), and **k** their mean fluorescence intensity quantification ($n = 3$). Error bars are presented as mean ± standard deviation (SD). The data were analyzed by one-way ANOVA with Tukey's post hoc test. The experiments for (**b, c, f, i,** and **j**) were repeated three times independently with similar results. Source data are provided as a Source Data file.

isolated into several apoptotic bodies, the structure of the membrane is still intact without contents exudation[27].

To further explore irradiation-induced cell membrane damage-related death, the lipid peroxidation probe C11-BODIPY was used. With the increase of radiation dose, raising oxidized C11-BODIPY fluorescence was found (Fig. 2i–k), which suggests that the irradiation-induced membrane damage is due to lipid peroxidation. This violent phenomenon occurred after irradiation-induced ROS production (Supplementary Fig. 2a, b), indicating ROS directly increases lipid peroxidation to promote ferroptosis. Similar to the previous study, intracellular levels of ferroptosis-related genes were altered after ROS production by irradiation, and the administration of a ROS scavenger (N-acetylcysteine) and Fer-1 had similar effects on the recovery of cell survival from irradiation injury[21]. ROS generated by irradiation can extract electrons from PUFAs to form PUFA radicals (PUFA˙). Then, these unstable radicals rapidly interact with oxygen molecules to generate lipid peroxyl radicals (PUFA-OO˙), extract H˙ from other molecules through the Fenton reaction, and produce lipid hydrogen peroxide (PUFA-OOH) in the end[3]. Therefore, ferroptosis has a crucial role in irradiation-induced intestinal injury. Thus, scavenging ROS and inhibiting ferroptosis are expected to be effective for radioprotection.

## CeO$_2$@HNTs@DFP nanoformulations with ROS scavenging and ferroptosis inhibition

In this work, an oral formulation was designed to inhibit ferroptosis for the treatment of radiation colitis. Clay mineral HNTs were used as a functional carrier, and CeO$_2$@HNTs were synthesized by in situ growth of CeO$_2$ on the outer surfaces of the tubes. Afterward, CeO$_2$@HNTs@DFP was prepared by loading deferiprone (which can chelate free or protein-bound iron (III) to inhibit ferroptosis). Finally, CeO$_2$@HNTs@DFP stabilized Pickering emulsion was prepared by adding VE oil phase emulsification, and the emulsion can be stored stably for more than one week (Fig. 3a).

The in situ growth of CeO$_2$ on HNTs was achieved by a modified co-precipitation method in an alkaline medium through the hydrolysis of Ce$^{3+}$ to form Ce(OH)$_3$, which was gradually oxidized by airborne O$_2$[11]. The outer surface of HNTs is composed of silica-oxygen tetrahedra, which are strongly negatively charged in alkaline media and can provide active sites for Ce$^{3+}$ by electrostatic adsorption, followed by nucleation and growth[15]. Briefly, ethylene glycol solution of Ce(NO$_3$)$_3$·6H$_2$O was added slowly to the aqueous dispersion of HNTs and gradually heated to 60 °C, then ammonia (28–30%, precipitant) was added. Ethylene glycol has a high viscosity and can adsorb on the surface of CeO$_2$ nuclei to inhibit their growth into large sizes.

The morphology of CeO$_2$@HNTs was analyzed by transmission electron microscope (TEM) (Fig. 3b). Compared with raw CeO$_2$, CeO$_2$@HNTs have less CeO$_2$ agglomeration and the CeO$_2$ nanoparticles are uniformly dispersed on the surfaces of HNTs. The size of CeO$_2$ decreased from 4.7 nm and 2.8 nm in the CeO$_2$@HNTs sample. The possible reason is that the presence of the Si–O–Si and metal ions interfaces significantly reduces the nucleation barrier, and thus heterogeneous nucleation of CeO$_2$ occurs at the interface[28]. The elemental mapping shows that the Ce atom is well distributed on HNTs (represented by Al, Si) (Fig. 3c). X-ray diffraction (XRD) patterns (Fig. 3d) show that the prepared CeO$_2$ has distinct featured diffraction peaks, and the peaks of CeO$_2$ in CeO$_2$@HNTs are broadened due to the decreased particle size. The peaks at 28.7°, 33.1°, 47.6°, and 56.5° correspond to [111], [200], [220], and [311] planes of CeO$_2$, respectively[29]. The particle size of the nanoparticles can be calculated from the Scheller equation to be about 2.9 nm, which agrees with the average particle size obtained in the TEM. The particle size affects the catalytic activity since smaller nanoparticles have a higher surface area to volume ratio. The reduced particle size and improved dispersion contribute to the catalytic activity of the CeO$_2$ nanozyme.

Nanozyme performs catalytic enzyme-like functions dependent on the presence of two valence states on a single element, Ce in ceria can shift readily between Ce$^{3+}$ and Ce$^{4+}$ oxidation states. Ce$^{3+}$ sites perform SOD-like functions by removing ˙O$_2^-$ and •OH via redox reactions, while Ce$^{4+}$ sites perform CAT-like functions by oxidizing H$_2$O$_2$[30]. A high percentage of Ce$^{3+}$ in CeO$_2$ nanozyme is vital for the treatment of inflammation-related diseases because ˙O$_2^-$ and •OH are directly associated with inflammatory responses and cell death[30]. X-ray photoelectron spectroscopy (XPS) showed (Fig. 3e, Supplementary Fig. 5, and Supplementary Table 1) that the addition of HNTs increased the percentage of Ce$^{3+}$ (from 20.7% of free CeO$_2$ to 28.2% of CeO$_2$@HNTs). This phenomenon may be due to the smaller particle size and higher dispersion of CeO$_2$ generated on the silica surface of HNTs, conferring more oxygen vacancies[31]. The content of CeO$_2$ and Ce, determined by thermogravimetric analysis (TGA) (Fig. 3f) and inductively coupled plasma-optical emission spectrometry (ICP-OES) (Supplementary Fig. 6), was 28.6% (CeO$_2$) and 21.1% (Ce atoms), respectively. The amounts of materials in subsequent studies were normalized by the Ce content.

Three representative ROS, including ˙O$_2^-$, H$_2$O$_2$, and •OH, were selected to evaluate the catalytic performance (Fig. 3g–i). The higher SOD-like activity shown by CeO$_2$@HNTs in Fig. 3g and Supplementary Fig. 7a can be attributed to the fact that ˙O$_2^-$ scavenging is mainly controlled by Ce$^{3+}$ status, and the increased Ce$^{3+}$ ratio of CeO$_2$ on the surface of HNTs favors the exertion of SOD-like activity. H$_2$O$_2$ originates from the disproportionation of ˙O$_2^-$, and excess H$_2$O$_2$ is a major cause of inflammation[32]. CAT-like activity decomposes into H$_2$O and O$_2$ and can be evaluated by monitoring the O$_2$ produced. CeO$_2$@HNTs (at Ce concentration of 100 μg mL$^{-1}$) produced slightly more O$_2$ than CeO$_2$ (not significant), but interestingly, when the concentration continued to increase, the CAT-like activity of CeO$_2$@HNTs started to be significantly better than that of CeO$_2$ (Supplementary Fig. 7b, c). The reason may be that the strong positive charge of CeO$_2$ makes it easy to agglomerate in water at high concentrations (Supplementary Fig. 8a), while the increased negative charge of HNTs doping (Supplementary Fig. 8b) improves the water dispersibility and maintains the catalytic activity of CeO$_2$. In addition, the electron paramagnetic resonance (EPR) spin capture technique was used to evaluate the •OH scavenging function. •OH was generated by the Fenton reaction with the Fe$^{2+}$/H$_2$O$_2$ system and was captured by 5,5'-dimethylpyrroline-1-oxide (DMPO)[33]. The EPR spectroscopy results illustrated that the intensity of the DMPO-OH peak was significantly lower in each group of CeO$_2$ synthesized by HNTs (Fig. 3i). The CeO$_2$@HNTs exhibit excellent ROS scavenging performance in all three assays, suggesting that the smaller particle size and higher dispersion of CeO$_2$ generated on the silica surface of HNTs greatly enhanced the ROS scavenging activity. Cell viability tests showed that CeO$_2$@HNTs were biocompatible at appropriate concentrations, indicating that their strong catalytic activity does not affect the natural physiological processes of intestinal epithelial cells (Supplementary Fig. 9). It is promising for scavenging irradiation-generated ROS to alleviate inflammation.

CeO$_2$@HNTs@DFP was obtained by loading DFP into the tubes by cyclic vacuum pumping. The lumen diameter of 16.9 ± 3.4 nm (Supplementary Fig. 10) of HNTs provides a high capillary force to push the drug solution into the tube, supplemented by cyclic pumping for the highest in-tube loading[34]. The DFP encapsulated on CeO$_2$@HNTs can be identified by the characteristic absorption bands of the Fourier-transform infrared spectroscopy (FTTR) spectrum (Supplementary Fig. 11a). The peaks around 3139 cm$^{-1}$ (-OH), 1630 cm$^{-1}$ (C=O, ketone), 1564 cm$^{-1}$ (C–N), 1514 cm$^{-1}$ (C=C, aromatic), and 1463 cm$^{-1}$ (CH$_3$) were assigned to the DFP. Further, TGA curves (Supplementary Fig. 11b) show the DFP loading up to 10.9 wt% of the HNTs weight. The drug almost completely filled the lumen since the density of DFP and HNTs is 1.23 g cm$^{-3}$ and 2.53 g cm$^{-3}$, respectively. From Supplementary Fig. 11c, CeO$_2$@HNTs@DFP gradually releases DFP over 2–12 h. The

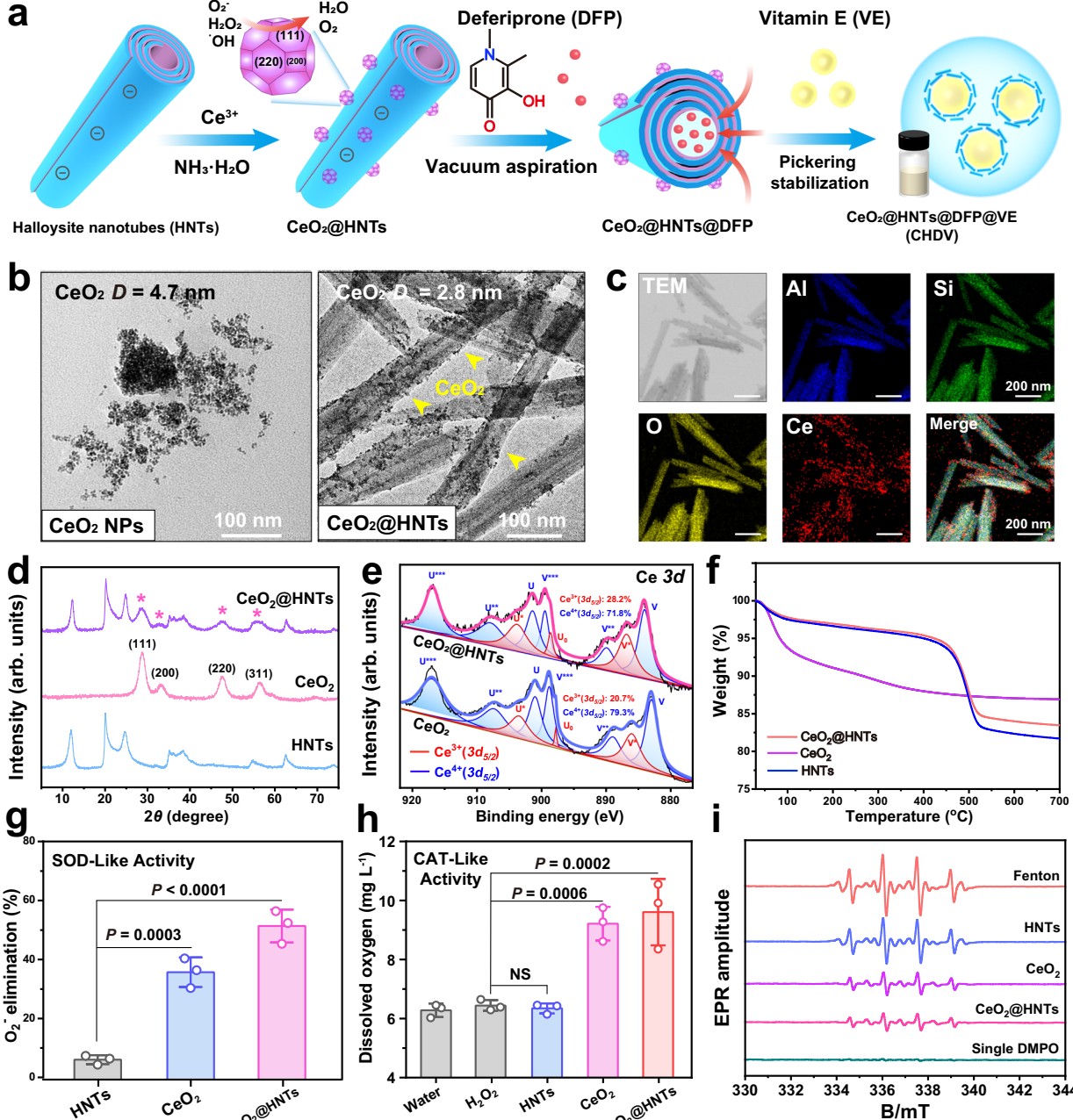

**Fig. 3 | Synthesis and characterization of CeO₂@HNTs. a** Schematic illustration of CHDV Pickering emulsion. **b** TEM images of CeO₂ and CeO₂@HNTs, scale bars, 100 nm. **c** TEM image and elemental mapping of CeO₂@HNTs, scale bars, 200 nm. **d** XRD patterns, **e** Ce 3d XPS spectra, and **f** TGA cruces of HNTs, CeO₂, and CeO₂@HNTs. **g–i** Enzyme-like activity tests of HNTs, CeO₂, and CeO₂@HNTs, the amounts of materials were normalized by Ce content (100 μg mL⁻¹). **g** ˙O₂⁻ elimination rate by SOD-like activity, **h** oxygen generation from H₂O₂ catalyzed by CAT-like activity, and **i** EPR spectra of the •OH scavenging ability of materials using DMPO as a spin trap agent. Error bars are presented as means ± SD, n = 3 independent repeats. The data were analyzed by one-way ANOVA with Tukey's post hoc test. NS means no significant difference. The experiments for (**b**, **c**) were repeated three times independently with similar results. Source data are provided as a Source Data file.

release profile conforms to the Korsmeyer–Peppas model ($M_t/M_\infty = kt^n$, where $M_t$ is the amount of DFP released at time $t$, $M_\infty$ is the amount of drug released at infinite time, $n$ is the exponential characteristic mechanism of release, and $k$ is a constant)[35]. Within 12 h, $n \leq 0.45$, indicating that the mechanism of DFP release is Fickian diffusion. As evidenced by XRD reflections, the DFP does not intercalate between the layers of HNTs, but only is loaded in the lumen (Supplementary Fig. 11d).

For the long-term stability of the oral nanoplatform, CeO₂@HNTs@DFP was prepared as CeO₂@HNTs@DFP@VE (CHDV)

Pickering emulsions. CHDV exhibited potent stability, with constant particle size from top to bottom layer of the emulsion droplets after 7 days of placement, thanks to their high resistance to agglomeration (Supplementary Fig. 12). HNTs play a major role in forming of Pickering emulsions, and FITC-labeled HNTs can be observed in the outer ring of Nile Red-stained emulsion droplets (Supplementary Fig. 13). The increase in HNTs content can form a thicker protective shell at the oil-water interface, resulting in smaller droplet sizes and more stable emulsions[36]. In addition, the lipid involvement in the emulsion increases the lipophilicity of the nanoparticles, thus increasing the

affinity for the cell membrane of intestinal epithelial cells (Supplementary Fig. 14).

Olive oil containing VE (0.1 g mL$^{-1}$) and aqueous dispersions of CeO$_2$@HNTs@DFP were used as oil and water phases to make CHDV through sonication and homogenization. Oleic acid (the main component of olive oil) has a partition coefficient similar to that of phospholipid bilayers, resulting in rapid binding to the cell membrane with high affinity[37]. VE, on the other hand, has been used as a radioprotective agent in radiotherapy (and has recently been proven to inhibit ferroptosis) by effectively "trapping" peroxyl radicals, preventing lipid oxidation, and inhibiting lipoxygenases[38]. Enzyme-like activity tests indicate that CHDV maintains similar ROS scavenging properties to CeO$_2$@HNTs (Supplementary Fig. 15). To summarize, CHDV emulsions have a combination of oral stability, enhanced cell membrane affinity, and intact enzyme-like activity.

## CHDV Pickering emulsion preferentially adheres to inflamed mucosa

The strong negative charge conferred by HNTs contributes to the passive deposition of CHDV into the positively charged inflamed colon. A characteristic feature of irradiation-induced intestinal injury is the thinning or loss of the mucus layer due to damage to the mucus-producing goblet cells[39]. Infiltration of intestinal contents at the mucus defect sites further induces inflammation and in situ accumulation of positively charged proteins, including transferrin, bactericidal/permeability proteins, and antimicrobial peptides[40]. These pathophysiological features of mucosal defect-inflammation co-localization and local charge alteration allow negatively charged nanoparticles to target intestinal inflammation by electrostatic interactions (Fig. 4a)[41].

CHDV Pickering emulsions have a tiny particle size (9.8 μm on average) (Fig. 4b–d), which makes them susceptible to electrostatic attraction and capture by cells. Figure 4e shows that the potential of CHDV at pH 3 (gastric acid) to pH 8 (colon fluid) decreases from −17.1 ± 0.1 to −27.3 ± 0.1 mV. Taking the strong positive charge of CeO$_2$, the negative charge of CHDV is endowed by HNTs interposed at the oil-water interface (Fig. 4d, e). Due to the hollow tubular structure, the positively charged AlO$_2$(OH)$_4$ layer of HNTs is located in the interior, giving it the strongest negative charge among various clays (e.g., montmorillonite, kaolin, diatomite)[13]. Next, CHDV was incubated on polystyrene plates coated with mucin (negatively charged) or transferrin (positively charged) to simulate the electrostatic interactions between them and the surfaces of healthy and inflamed colons, respectively (Fig. 4f). It was observed that the transferrin-coated plates retained sevenfold more fluorescent signal from CHDV than the mucin-coated plates after washing.

The ex vivo and in vivo adhesion tests of CHDV to inflamed colon tissue were examined using radiation colitis mice. After ex vivo incubation of CHDV with healthy or colitis colon segments, fluorescence imaging photographs revealed significantly higher fluorescence retention in colitis mice's colon than in healthy controls (Fig. 4g). For in vivo testing, mice were orally administered with CHDV, and after 10 h, fluorescence imaging and quantitative analysis of the entire intestine revealed significantly increased adhesion and retention of CHDV in colitis mice's colon (Fig. 4h). Scanning electron microscope (SEM) images showed that CHDV (traced by characteristic tubular HNTs) was enriched in mucosal defects and rare in sites with intact mucus gel (Fig. 4i). Similarly, as shown in the fluorescence imaging of the colonic cross-sections in Fig. 4j, the colonic tissues of colitis mice exhibited blurred crypt borders due to injury, and there was a significant enrichment of CHDV fluorescence in the damaged (inflamed) areas.

## In vitro ferroptosis reversal and evaluation of radiation protection

To investigate the therapeutic effect of CHDV inhibition of ferroptosis on radiation colitis, ferroptosis was induced in intestinal epithelial cells using the ferroptosis inducer (RAS-selective lethal 3, RSL3) (IC$_{50}$ = 0.8 μM) (Supplementary Fig. 16). RSL3 specifically inhibits ferroptosis without bothering other cell death pathways. RSL3 directly binds to glutathione peroxidase 4 (GPX4) to inactivate it, induces intracellular physiological lipid ROS production, and thus induces ferroptosis[42]. CHDV was non-cytotoxic to intestinal epithelial cells without irradiation or RSL3 exposure (Supplementary Fig. 17). When exposed to RSL3 (0.8 μM), cell viability rebounded with increasing concentrations of CHDV, indicating that CHDV inhibited ferroptosis in intestinal epithelial cells (Fig. 5a). CHDV also showed significant protection under gradient RSL3-induced ferroptosis at different levels (Fig. 5b). This inhibitory effect on ferroptosis is mainly achieved by the release of DFP (Supplementary Fig. 18), which can restrain ferroptosis by chelating iron necessary for the catalysis of lipoxygenase[43].

The protective effect is achieved by alleviating lipid peroxidation, which can be demonstrated by monitoring the fluorescence intensity of oxidized C11-BODIPY. CHDV significantly reduces RSL3 and irradiation-induced lipid peroxidation (Fig. 5c and Supplementary Fig. 3a, b). Similarly, fluorescence images of cells displayed the same trend after CHDV applying; the reduced form of C11-BODIPY (red) increased, whereas the oxidized form of C11-BODIPY (green) decreased accordingly (Fig. 5d). Cellular calcein-AM/propidium iodide (PI) staining revealed that the number of PI-accessible cells (cells with damaged membranes) was significantly reduced in CHDV groups (Fig. 5e). These results suggest that DFP released by CHDV blocks the lipid peroxidation process in intestinal epithelial cells, which preserves the integrity of the cell membrane and thus avoids cell death. Iron chelators have been found to alleviate radiotherapy-induced fibrosis[44], muscle[45] and bone damage[46] by reducing ROS in cells and promoting vascular renewal.

Synergistic treatment of ROS scavenging and ferroptosis inhibition can alleviate irradiation injury in intestinal epithelial cells. It was found that CHDV treatment also decreased RSL3 and radiation-induced ROS accumulation in intestinal epithelial cells by monitoring the ROS probe (2′,7′-Dichlorofluorescein diacetate, DCFH-DA) (Supplementary Figs. 3d, e and 19). Subsequently, the cell viability and proliferative capacity were assessed after X-ray exposure. Colony formation assays (Fig. 5f) showed that irradiation severely blocked enterocyte proliferation and growth, whereas cells were gradually protected with increasing CHDV concentrations. Colony formation tests were performed after splitting the antioxidant and ferroptosis-inhibiting components of CHDV (Supplementary Fig. 20). The results showed that both therapeutic mechanisms contributed to protecting irradiation-induced cell damage. Similar to DFP, Fer−1 alone also can slightly reverse the irradiation damage, but not as effective as CHDV (Supplementary Fig. 3f, g). Irradiation-induced DNA double-strand breaks were also reduced (Supplementary Fig. 21), which could be attributed to the scavenging of ROS by CeO$_2$[47]. The VE in the oil phase of the emulsion also plays a certain role in neutralizing the ROS generated by radiation exposure by donating H atoms[48]. Together, these results suggest that CHDV effectively reduces ferroptosis in irradiated cells by blocking lipid peroxidation and scavenging ROS, efficiently rescuing irradiation-induced cell death.

## In vivo therapeutic efficacy

Encouraged by the excellent inflammatory targeting and radio-protective properties of CHDV, in vivo radiation colitis protection was further evaluated in C57BL/6J mice. First, the oral safety of CHDV was evaluated. The body weight of the mice increased normally as in the control group during 7 days of continuous oral administration of CHDV (Supplementary Fig. 22). Blood biochemical and blood routine tests in mice on day 7 showed no significant difference between the CHDV group and the control group as well (Supplementary Fig. 23). These results demonstrate that CHDV meets the safety requirements of the animal model. After gavage to mice, CHDV was most enriched in

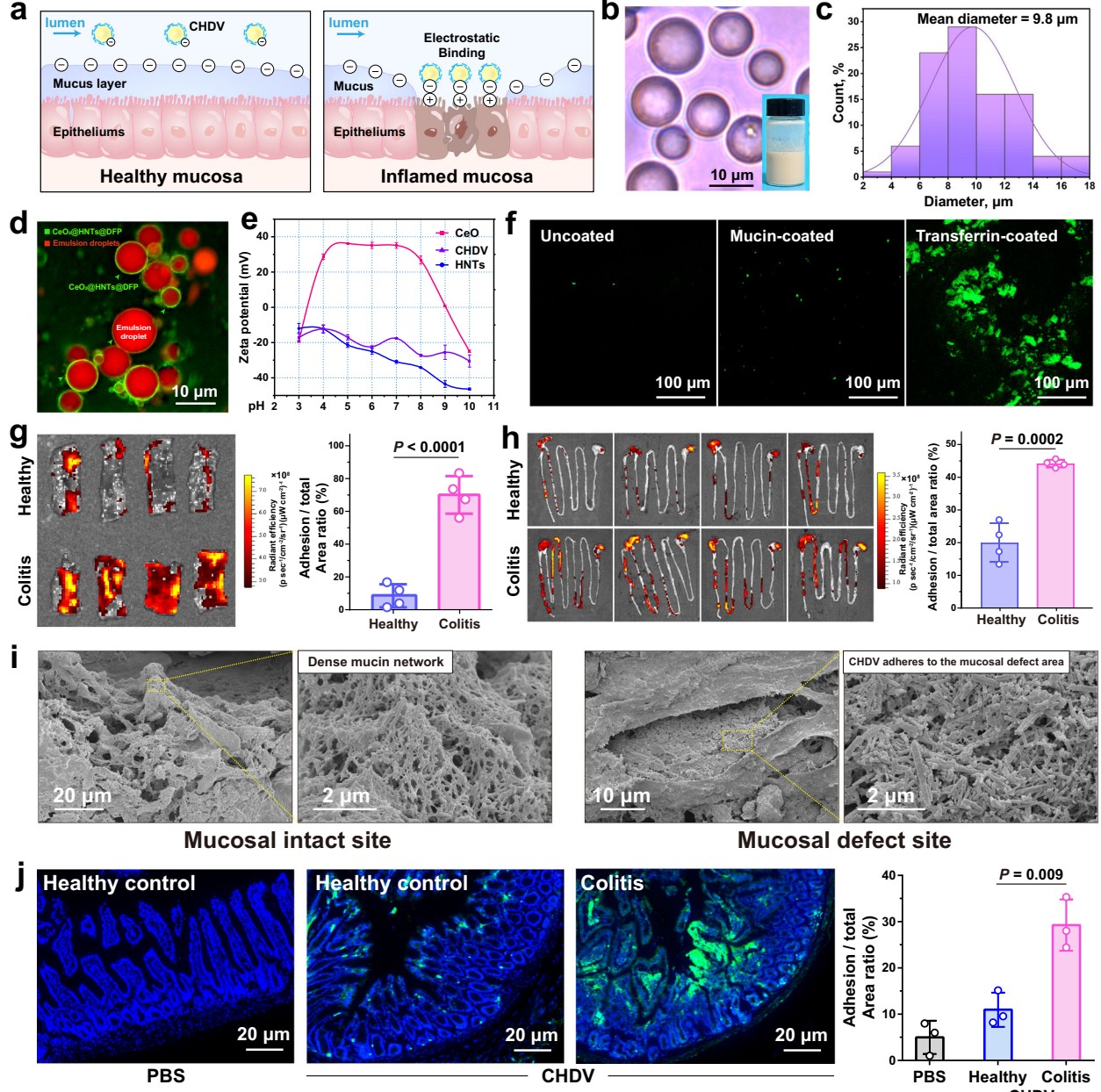

**Fig. 4 | CHDV Pickering emulsion preferentially adheres to inflamed mucosa.** **a** Schematic diagram of healthy and inflamed mucosa. The latter is characterized by mucosal defects and the accumulation of positively charged proteins. **b** The microscope images (scale bars, 10 μm), **c** corresponding particle size distribution statistics, and **d** fluorescence micrographs of CHDV, scale bars, 10 μm. **e** Zeta potentials of HNTs, $CeO_2$, and CHDV at different solution pH ($n = 3$). **f** CHDV was incubated with uncoated, mucin-coated (simulating healthy epithelium) or transferrin-coated (simulating inflamed epithelium) surfaces. Scale bars, 100 μm. **g** The distal colon of mice with radiation colitis and healthy control was incubated ex vivo with CHDV, and fluorescence was quantified by an IVIS imaging system ($n = 4$). **h** The mice were sacrificed 8 h after administration, and the intestines were dissected and imaged using an IVIS imaging system. The fluorescence images above were quantified by ImageJ software, and the percentage of CHDV retained in the intestine versus the whole intestine was quantified ($n = 4$). **i** SEM images of colitis mice mucosa after treatment with CHDV. **j** The fluorescent images of healthy and colitis mice colon sections after treatment with orally administered CHDV (scale bars, 20 μm), and the percentage of retained material versus the colon was quantified ($n = 3$). Error bars are presented as mean ± SD. $P$ values were calculated using a two-tailed unpaired Student $t$ test. The experiments for (**b**, **d**, **f**, **i**, **j**) were repeated three times independently with similar results. The experiments for (**g**, **h**) were repeated four times independently with similar results. Source data are provided as a Source Data file.

the colon at 6–8th hour, and there was no obvious distribution in other organs (Supplementary Fig. 24). The tubular structure of HNTs remained intact in all intestinal segments, indicating their gastrointestinal stability (Supplementary Fig. 25). According to biodistribution of CHDV, a field size of 1 cm² in the abdominal of irradiated mice was exposed to 15 Gy of X-ray at 8 h after oral gavage. Then continued daily dosing for 7 days, and perform hematochezia and histopathological were performed at the indicated time points (Fig. 6a).

To understand the involvement of ferroptosis in irradiation-induced intestinal toxicity, H&E staining was used to evaluate the histopathological outcome of the modeling method. As displayed in Supplementary Fig. 26, the colonic area showed shortened villi, localized absence of crypts, massive immune cell infiltration, and localized microhemorrhage, exhibiting colitis features. Due to localized pelvic radiation avoiding systemic exposure, the small intestinal villus length and crypt integrity were in the normal range, even in the proximal

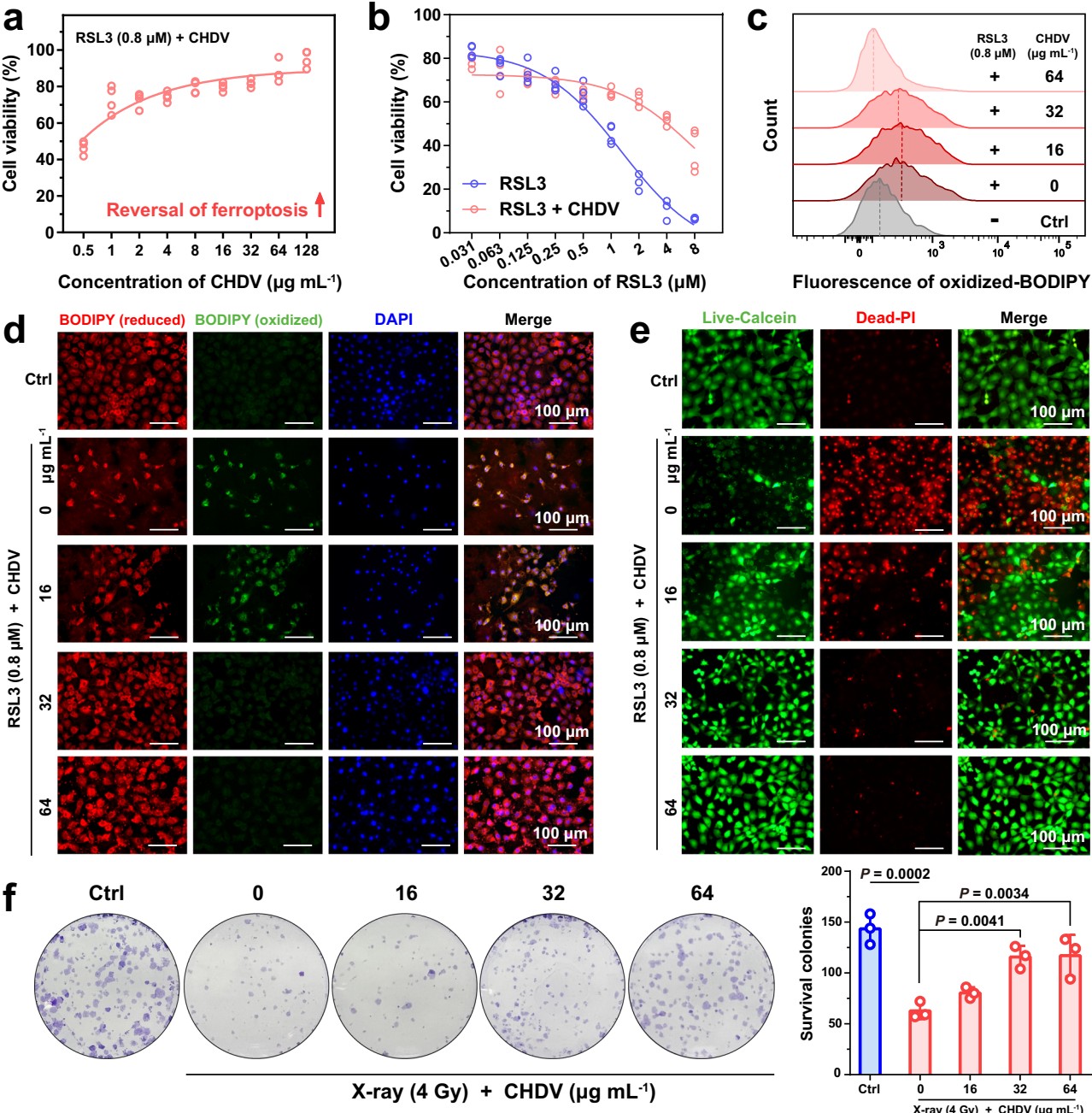

**Fig. 5 | In vitro ferroptosis reversal and radiation protection. a** CDHV reverses RSL3 (0.8 μM)-induced ferroptosis in intestinal epithelial cells (*n* = 4). **b** CHDV (64 mg mL⁻¹) reversed various degrees of ferroptosis induced by gradient concentrations of RSL3 (*n* = 4). **c** The fluorescence intensity of ferroptosis-dependent C11-BODIPY oxidation decreased with increasing CHDV (10,000 cells per tube were collected). **d** The fluorescence images and **e** Calcein-AM/PI staining of cells after rescue by CHDV, scale bars, 100 μm. **f** Crystal violet staining and quantification of the surviving colonies irradiated with 4 Gy X-ray and different doses of CHDV treatment (*n* = 3). Error bars are presented as mean ± SD. The data were analyzed by one-way ANOVA with Tukey's post hoc test. The experiments for (**d**–**f**) were repeated three times independently with similar results. Source data are provided as a Source Data file.

small intestinal. As indicated in Fig. 6b, the body weight of the irradiation group drastically decreases compared with the control group. Surprisingly, CHDV treatment slightly increases the body weight compared to the irradiation group, whereas HNTs, CeO₂, DFP, or VE treatment showed a mild decrease or no significant change. Radiation exposure doses that injure the gastrointestinal tract (15 Gy) will also destroy the hematopoietic system, causing both acute and chronic oxidative injury[49]. Therefore, the survival rate of the mice in the irradiation group and HNTs group decreased sharply after irradiation, and all were dead on the 11th and 16th day, respectively, indicating the severity of pelvic radiotherapy without antioxidative or cell death-

suppressing treatment. CeO₂, DFP, or VE alone treatment shows a survival rate of 10%, 20%, and 20%, respectively, which indicates that they could not sufficiently protect the irradiated mice against irradiation when used separately (Fig. 6c). This may be due to imprecise medication leading to insufficient drug availability at the damaged site and the pointless antioxidant therapy that cannot rescue irradiation-induced multiple cell death pathways. However, mice administrated with CHDV exhibited a survival rate of 60% up to 30 days, preliminarily implying the notable protective effect of CHDV on radiation colitis.

The radiation colitis model mice all developed symptoms of colitis, including diarrhea (wet tail) and hematochezia (fecal occult

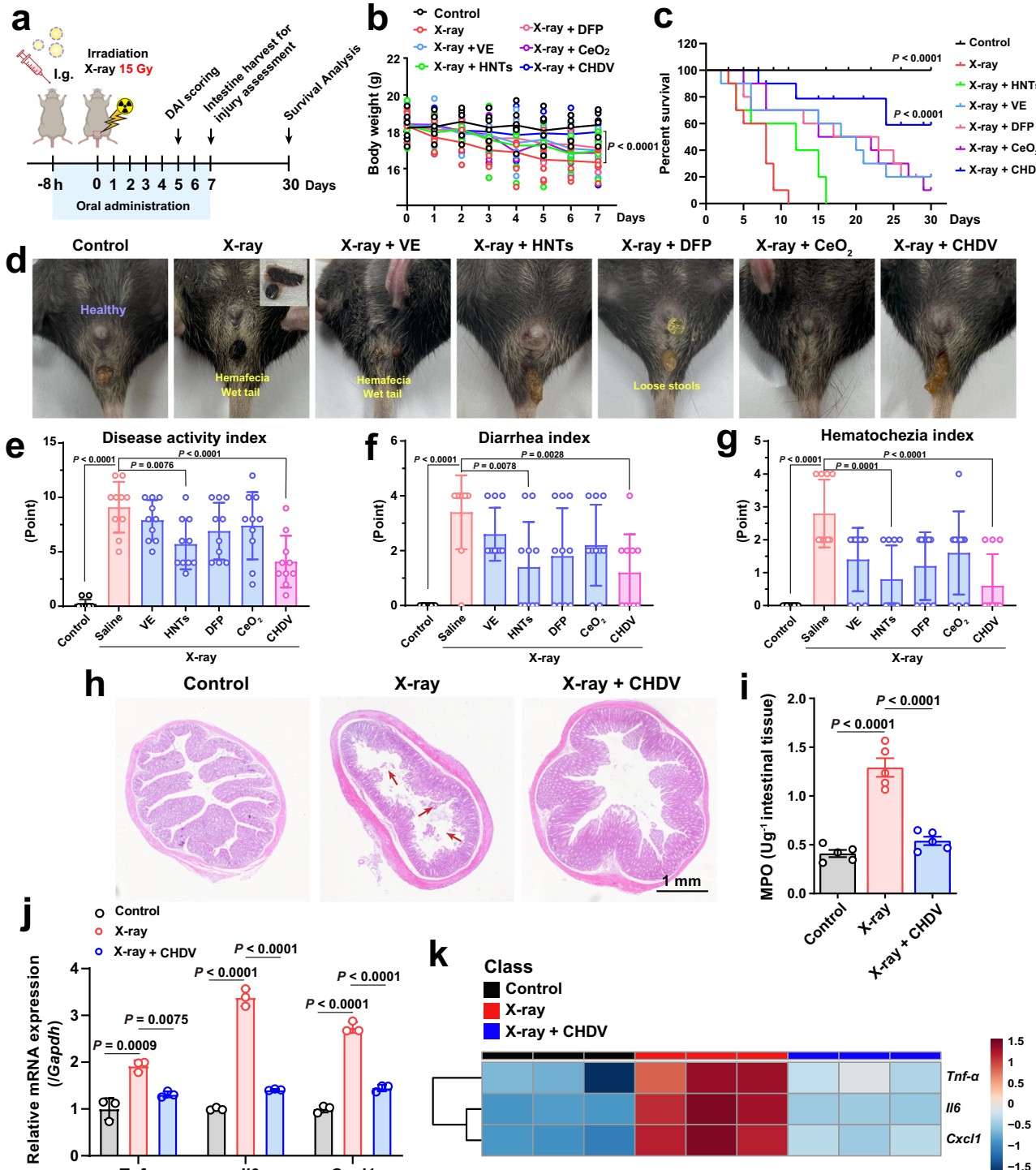

**Fig. 6 | Radioprotection of CHDV alleviates radiation colitis. a** Schematic illustration for in vivo radioprotection evaluation. **b** Body weight of mice ($n = 6$). **c** Survival curves at day 30 ($n = 10$), the data were analyzed by Log-rank test. $P < 0.0001$ vs the X-ray group. **d** Representative photographs of the wet tail and hematochezia on day 7 in various groups. **e** DAI of radiation colitis, **f** diarrhea index, and **g** hematochezia index in each group ($n = 10$). **h** Representative photographs of H&E staining of colon tissue after 7 days of treatment. Scale bars, 1 mm. The experiment was repeated six times independently with similar results. **i** The activity of MPO in colon tissue ($n = 5$). **j** The gene expressions of *Cxcl1*, *Tnf-α*, and *Il6* in colon tissue were detected by the q-PCR assay ($n = 3$), and **k** they were quantified and visualized by heatmap. The data are represented as mean ± SD. The data were analyzed by one-way ANOVA with Tukey's post hoc test. Source data are provided as a Source Data file.

blood) (Fig. 6d), mainly as a result of rapidly proliferating enterocyte death and an acute inflammatory response in the lamina propria[1]. These symptoms were scored by the disease activity index (DAI) in each group (Fig. 6e). The results revealed that diarrhea and hematochezia were relieved to a certain extent in the HNTs-containing

groups, partially relieved in the CeO₂, DFP, and VE groups, while the CHDV group significantly suppressed diarrhea and hematochezia (Fig. 6f, g). These data indicate that HNTs-based administration is very effective in the anesis of diarrhea and hematochezia caused by colitis. These results are consistent with previous reports on the efficacy of

clays in ulcerative colitis. They believed that negatively charged kaolin clay adheres to the damaged mucosa by electrostatic interaction, adsorbs toxic substances and pathogens, avoids further mucosal damage, and regulates gut microflora[13]. Causes of bleeding in radiation colitis include irradiation-related vasodilation, mucosal edema, and ulcers[50]. Recently, hemostatic powders based on inorganic minerals have been approved for endoscopy of radiation colitis. They work by blocking active bleeding sites, absorbing blood, and then acting as a bandage to stop bleeding[51]. This therapy coincides with the previous reports of HNTs as a hemostatic agent, which stops bleeding by absorbing water to concentrate the blood and activate platelets and clotting pathways[14].

The protection of irradiation-induced colon injury by CHDV was further confirmed by H&E staining. The pathologic injury was indicated by the red arrows in the colon tissue. As depicted in Fig. 6h, the colon in the irradiation group induced severe destruction of the villi structure. Conversely, CHDV treatment remarkably protected colon mucosa, and the villi were less injured by irradiation and remained virtually intact. Although the exact pathogenesis of radiation colitis remains unclear, immune and inflammatory responses are definitely involved and play essential roles[52]. The anti-inflammatory effect of CHDV was further investigated in irradiated mice. As presented in Fig. 6i–k, irradiation can induce strong inflammatory responses, accompanied by increasing activity of myeloperoxidase (MPO) and promoting the expression of pro-inflammatory cytokines in colon tissue, such as chemokine ligand 1 (*Cxcl1*), tumor necrosis factor (*Tnf-α*) and interleukin-6 (*Il6*). On the contrary, CHDV treatment obviously attenuated the expression of pro-inflammatory cytokines and reduced MPO activity. Combined with tissue sections and inflammatory indexes, CHDV effectively mitigates irradiation-induced colon injury and immunoreaction. H&E staining also indicated no significant damage to major organs associated with CHDV and its individual components, suggesting that their efficacy is not accompanied by organismal impairment (Supplementary Fig. 27).

It is known that the scavenging of ROS and the mitigation of oxidative stress can reduce the secretion of pro-inflammatory cytokines and protect impaired colon tissue[10]. Activation of inflammatory signals at the injury site induces the recruitment and activation of immune cells (e.g., macrophages). Significantly elevated levels of inflammatory factors in radiation colitis lesions induce macrophages and neutrophils from the circulation to the inflamed mucosa[53]. The lipid peroxidation products produced by ferroptosis also recruit macrophages[6]. In addition, the lipid peroxidation product 4-HNE serves as a pro-inflammatory mediator that triggers the NF-κB pathway, also recruiting macrophages[54].

To verify whether CHDV exerts its effect through regulating the function of macrophages, immunofluorescence co-staining was performed in the radiation colitis tissue (Supplementary Fig. 28). Reduced recruited macrophages and less 4-HNE were observed in the CHDV treatment group, implying that the material inhibited ferroptosis and alleviated inflammation, thereby reducing macrophage recruitment. The anti-inflammatory function of CHDV may be partly derived from CeO₂ and VE[11]. These data demonstrate that CHDV could effectively reduce the inflammatory response against irradiation-induced colon injury and improves survival in irradiated mice. The translational dose range for CHDV is considered effective from 10 to 30 mg kg⁻¹ (adults) a day.

## Ferroptosis as a target for protection against radiation colitis

Disruption of intestinal tight junctions caused by enterocyte death and further infection and inflammation are important pathogenic factors in irradiation-induced colon injury[17]. This work demonstrated that multifactorial-induced ferroptosis, including ROS accumulation, lipid peroxidation, and iron overload from intestinal bleeding, is a critical pathway in irradiation-induced enterocyte death[19,20]. Therefore, the

alleviation of radiation colitis by CHDV was confirmed by inhibiting ferroptosis. Malondialdehyde (MDA), a decomposed product of lipid peroxides, can be used as a biomarker of lipid oxidation to assess ferroptosis levels[25]. MDA levels in the irradiated colon tissue were measured. As expected, compared with the control group, irradiated colon tissue in the irradiation group exhibited obviously higher MDA levels, while CHDV treatment remarkably reversed MDA levels in irradiated colon tissue (Fig. 7a). In addition, the histological analysis with Perls Prussian blue was performed to detect iron deposits (blue spots) in colon tissue. Increased iron deposits were found in irradiated colon tissue, possibly due to hemorrhage[55]. Decreased iron deposits were observed in the CHDV group (Fig. 7b), suggesting a remission of bleeding, which was attributed to the hemostatic activity of HNTs in CHDV. As total iron quantification illustrated in Fig. 7c, a significant increase of total iron content was discovered in the irradiated colon tissue, while CHDV treatment effectively reversed this phenomenon.

It is noted that CHDV exhibits an excellent ability to scavenge lipid oxides and chelate iron. Ferroptosis induction is involved in the increased expression of ferroptosis marker genes such as *Ptgs2*, *Acsl4*, *Ftl*, and *Fth1*[56]. Indeed, the X-ray irradiation-induced increase in colon *Ptgs2* gene expression was blocked by CHDV treatment (Fig. 7d, e). In order to confirm the impact of CHDV on inhibiting ferroptosis, the protein expression levels of several key biomarkers involved in ferroptosis pathways were examined in irradiated colon tissue. Interestingly, it was observed that both the gene and protein levels of ACSL4 were remarkably decreased after CHDV treatment in irradiated colon tissue (Fig. 7d–g), suggesting that the biosynthesis of PUFA-PL was not active. The lipid hydroperoxidase GPX4 is a crucial ferroptosis regulator that protects cells against membrane lipid peroxidation and maintains redox homeostasis. GPX4 was significantly depleted after radiation exposure, implying that radiation induces a surge of phospholipid hydroperoxides in membranes and lipoproteins in the colon. After treatment with CHDV, GPX4 levels were significantly restored, indicating that the ROS scavenging and antioxidant function of CHDV effectively reduced lipid peroxidation in the irradiated colon (Fig. 7d–g). Moreover, immunohistochemistry assays (Fig. 7h) further revealed that CHDV treatment strongly reduces ACSL4 and cytotoxic 4-HNE positive cells in irradiated colon tissue[57]. This result suggests that the radiotherapy-induced colonic lipid peroxidation process was reduced after oral administration of CHDV. This may be due to the scavenging of ROS by CeO₂ in CHDV, alleviating lipid oxidation stress.

In addition, compared with the irradiation group, the gene and protein levels of FTL and FTH were significantly decreased in the CHDV group. FTL and FTH are components of ferritin, which are essential in regulating iron metabolism. Their decline indicated that the iron uptake and storage in ferritin were blocked[58]. This phenomenon suggests that DFP in CHDV chelates large amounts of free iron in the colon, thereby relieving iron stress (partly due to hemorrhage). On the basis of the results described above, it was concluded that CHDV exhibits an excellent therapeutic effect on radiation colitis through inhibiting ferroptosis. For clinical purposes, fractionated radiotherapy is more commonly employed than the single high-dose radiotherapy used in this study to allow for the repair of normal tissues. However, it does not significantly alleviate the occurrence of ferroptosis. Overall, targeted and ferroptosis-inhibiting therapies hold the promise of minimizing normal tissue damage caused by radiotherapy.

Overall, by exploring lipid peroxidation and iron metabolism, the pathogenesis of radiation colitis from the perspective of ferroptosis was advanced. This discovery led to the development of an oral Pickering emulsion stabilized with nanoclay HNTs to alleviate radiation colitis by inhibiting ferroptosis. Specifically, ROS scavenging by CeO₂ grown in situ on HNTs, combined with intratubular loading of DFP to relieve iron stress, thereby rescuing ferroptosis in intestinal epithelial cells. Therein, HNTs play a critical role as either a medicinal clay to alleviate colitis, a nanocarrier that targets the inflamed colon by

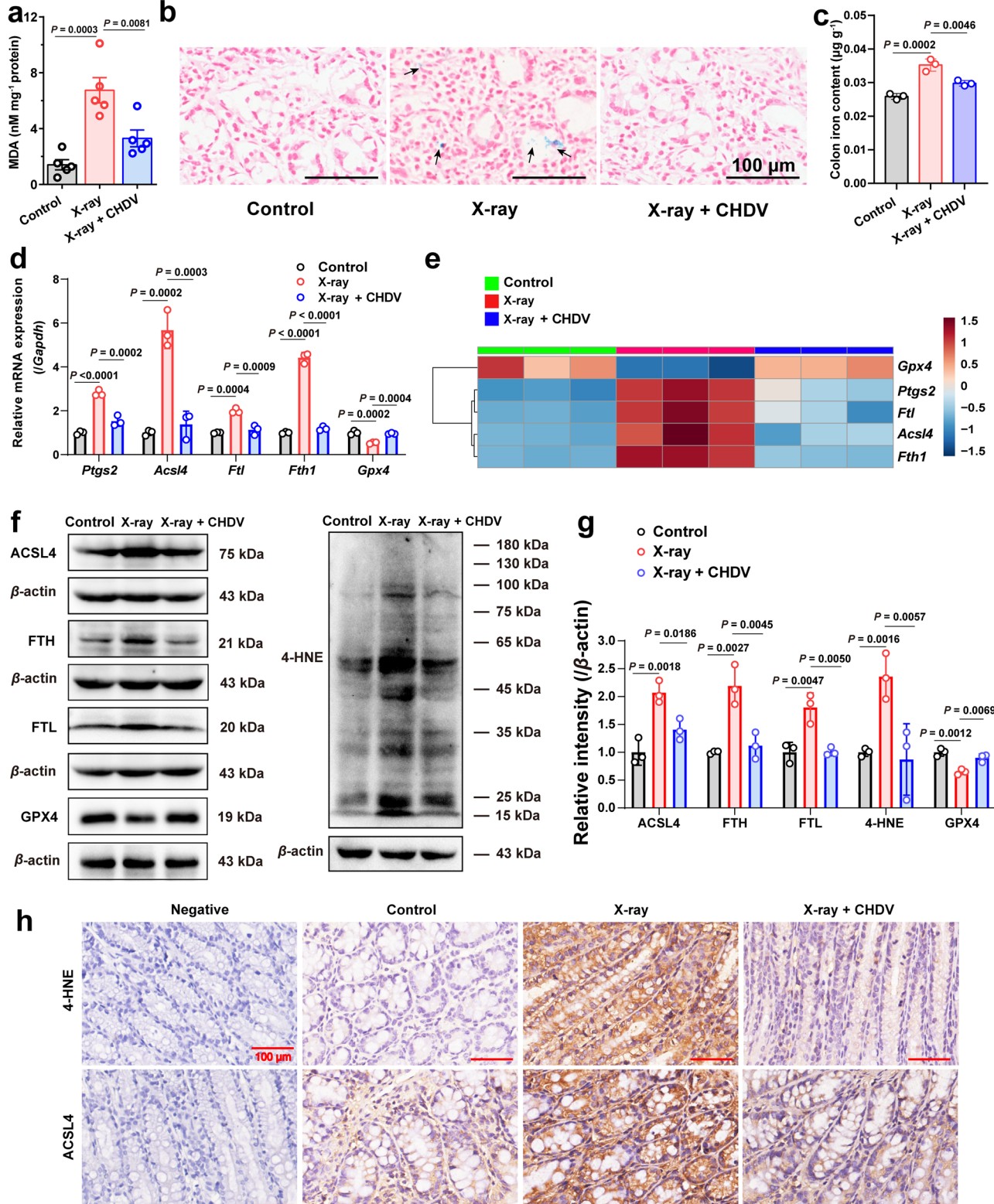

**Fig. 7 | CHDV protects against irradiation-induced colon injury by inhibiting ferroptosis. a** MDA levels in colon tissue (*n* = 5). **b** Representative images of colon sections detected by Perls Prussian blue staining. The blue spots (the black arrows) indicate the iron deposits in colon tissue. Scale bars, 50 μm. The experiment was repeated three times independently with similar results. **c** Histogram of colonic iron quantification analysis (*n* = 3). **d** The gene expressions of ferroptosis pivotal genes such as *Ptgs2*, *Acsl4*, *Ftl*, *Fth1*, and *Gpx4* in colon tissue (*n* = 3), and **e** they were

quantified and visualized by heatmap. **f**, **g** The protein expressions of ACSL4, FTL, FTH, 4-HNE, and GPX4 in colon tissue were determined by western blotting analysis (*n* = 3). **h** Immunohistochemistry of colon sections labeled with antibodies of 4-HNE (top row) and ACSL4 (bottom row) (*n* = 3). Scale bars, 100 μm. The data are represented as mean ± SD. The data were analyzed by one-way ANOVA with Tukey's post hoc test. The experiments for (**b, f, h**) were repeated three times independently with similar results. Source data are provided as a Source Data file.

electrostatic adsorption, or an interfacial stabilizer for oral Pickering emulsions. This ferroptosis-based synergistic strategy was effective in vitro and in vivo, providing pioneering insight into radiotherapy protection. Since ferroptosis and lipid peroxidation are involved in various diseases, this strategy proposed in this work provides a prospective insight for safe and effective treatment via rational regulation of oxidative stress.

## Methods

### Ethics statement

This retrospective study was approved by the Human Medical Ethics Committee of the Sixth Affiliated Hospital of Sun Yat-sen University (No.: 2023ZSLYEC-306). All research was conducted in accordance with relevant guidelines and regulations, and prior informed consent was obtained from all participants.

### Human samples collection

Colonic sections obtained from patients who underwent radiation colitis following radiotherapy between January 2021 and August 2021 were included in this study. Immunohistochemistry analysis was performed on a cohort of five patients with irradiated colon sites and their corresponding non-irradiated adjacent tissues. The protein expression intensity of immunohistochemistry was quantified using ImageJ software and normalized to the non-irradiated group. All patient data and samples were handled in strict accordance with patient confidentiality guidelines and in compliance with ethical standards. The patients involved in all the Cohorts were not part of any clinical trials, and all the samples were collected as a standard of care.

### Materials

High-purity HNTs were obtained from Guangzhou Runwo Materials Technology Co., Ltd., China. HNTs underwent multiple purifications to ensure high purity for biological research. Chemicals such as $Ce(NO_3)_3 \cdot 6H_2O$ (99.5%), ethylene glycol (AR, 98%) ammonia solution (AR, 25–28%), deferiprone were purchased from Macklin (China), and RSL3 were purchased from TargetMol (USA). All cell culture reagents were ordered from Thermo Fisher Scientific (USA). Molecular and cell biology reagents were purchased from the corresponding companies (marked when mentioned later).

### In situ synthesis of $CeO_2$ on HNTs

First, a uniformly dispersed suspension of 2 wt% HNTs was prepared. Then, 50 mL of ethylene glycol solution containing 630 mg $Ce(NO_3)_3 \cdot 6H_2O$ was added dropwise to 50 mL of HNTs dispersion. After vigorous stirring for 6 h, the mixture was heated at a rate of 5 °C min$^{-1}$, and when the temperature was raised to 80 °C, 8 mL of ammonia (28–30%) was injected into the mixture, and the reaction was kept at a constant temperature for 3 h. The product was washed by centrifugation using deionized water and ethanol until the pH of the supernatant was neutral. The product was lyophilized for further use.

### Preparation of $CeO_2$@HNTs@DFP

A saturated solution of DFP (100 μM) was prepared and then mixed with an aqueous dispersion of $CeO_2$@HNTs (0.1 wt%). The mixture undergoes cyclic vacuum pumping in/out with stirring, allowing the air to be expelled from HNTs lumen. The pumping was maintained for 0.5 h and exposed to air for 0.5 h. After three cycles, the product was washed by centrifugation several times with ultrapure water to remove excess DFP and then freeze-dried to obtain $CeO_2$@HNTs@DFP.

### Pickering emulsion preparation

$CeO_2$@HNTs@DFP aqueous dispersion (5 wt%) as water phase and VE dissolved in olive oil (0.1 g mL$^{-1}$) as oil phase. The water-oil mixture (in the ratio of 7:3) was sonicated and homogenized (13,000 × $g$) to obtain oil-in-water CHDV Pickering emulsion. To further confirm the

adsorption of $CeO_2$@HNTs@DFP at the oil-water interface, the oil phase was stained with Nile Red (530 nm excitation), and HNTs were labeled with FTIC (488 nm excitation), respectively. The microstructure of the emulsion droplets under the fluorescence field was captured by fluorescence microscopy (Axio Vert. A1, Zeiss, Germany).

### Characterization

The size, morphology, structure, and elemental distribution of $CeO_2$ and $CeO_2$@HNTs were analyzed using TEM (JEM−1400 Flash, JEOL Ltd., Japan) at an accelerating voltage of 120 kV. XRD patterns were obtained on an X-ray diffractometer (MiniFlex-600, Rigaku Corporation, Japan) with a scan rate of 5° min$^{-1}$ ranging from 5 to 80°. The element analysis was carried out by an XPS machine (ESCALAB250Xi, Thermo Fisher Scientific Ltd., USA). An FTIR spectrometer (PerkinElmer, UATR Two) was used to measure the FTIR spectrum, and the scan range was taken from 4000 to 400 cm$^{-1}$. The $CeO_2$ content and drug loading were analyzed by a TGA instrument (Mettler Toledo, Switzerland) from 30 to 700 °C at a heating rate of 10 °C min$^{-1}$ under an $N_2$ atmosphere. The content of Ce was measured using ICP-OES (Agilent 720, USA). The UV–vis spectra were measured by a UV–visible spectrophotometer (UV-2550, Shimadzu Instrument Ltd., Suzhou, China). The Zeta potentials of materials at different pH were analyzed using a Nano ZS zeta potential analyzer (Malvern Instruments Co., UK).

### SOD-like activity measurement

First, 20 μL of each material at a final concentration of Ce (100 μg mL$^{-1}$) was added to 200 μL of WST-1 (2-(4-iodophenyl)−3-(4-nitrophenyl)-5-(2,4-disulfophenyl)-2H tetrazolium sodium salt) working solution in the microplate wells, followed by 20 μL of enzyme working solution. This reaction produces a chromogenic reaction with superoxide anion. After incubation at 37 °C for 30 min, the absorbance of the samples at 450 nm was measured using a microplate reader (Bio-Tek, Hercules, USA). The color reduction meant a better superoxide elimination rate. Accuracy was ensured by three independent replicate experiments.

### CAT-like activity measurement

The CAT-like activity was assessed by measuring the oxygen production from $H_2O_2$ with a dissolved oxygen meter. Briefly, materials at a final concentration of Ce (100 μg mL$^{-1}$) were mixed with $H_2O_2$ (5 mM final concentration) in 5 mL of water. After 30 min of reaction at room temperature, the reaction solution was centrifuged to remove the materials, and the dissolved oxygen in the supernatant was detected using a dissolved oxygen meter (JPB-607A, INESA Scientific Instrument Co., Ltd, China). The higher the dissolved oxygen content, the stronger the CAT-like activity. Experiments were performed in three independent replicates.

### Scavenge •OH by EPR spectra analysis

The •OH-rich solution was generated by the Fenton reaction of the $Fe^{2+}/H_2O_2$ system (1.8 mM $FeSO_4$ and 5 mM $H_2O_2$). The •OH-rich solution was mixed with the materials (ensuring the final Ce concentration was 200 μg mL$^{-1}$) and incubated for another 30 min. After that, DMPO was added to the mixed solution for 10 min, and the ability of •OH radical scavenging was detected using an electron paramagnetic resonance (EPR, A300, Bruker, USA).

### Cell lines, cell culture, and cytotoxicity tests

Intestinal epithelial cell line IEC-6 was obtained from the American Type Culture Collection (ATCC, Manassas, Virginia, USA), which was cultured under the condition of a humidified atmosphere (5% $CO_2$) at the temperature of 37 °C. The complete medium for culture consisted of 90% Dulbeccos-modified eagle medium, 10% fetal bovine serum, and 1% penicillin-streptomycin. Cytotoxic effects of CHDV alone, X-ray

irradiation on the proliferation of cells, and RSL3-induced ferroptosis were examined by MTT assay or calcein-AM/PI staining according to the manufacturer's instructions. The operating parameters of X-ray irradiation were set as follows: 225 kV/13.33 mA high-energy X-ray, AP-PA technique, dose rate: 200 cGy min⁻¹ (X-Rad 225 XL, Precision inc., USA).

## Colony formation assay

Irradiated cells (0, 2, 4, or 6 Gy) that were treated or untreated with CHDV, CeO$_2$@HNTs + VE, or DFP were cultured into a 12-well plate at the density of 500 cells per well. For a comparative study of Fer-1 and CHDV, cells were seeded in a 6-well plate at the density of 1000 cells per well. After irradiation, cells were incubated for 10 days to form cell colonies. The cell colonies were stained with 1% crystal violet dye (SL7081, Coolaber, China), and the number of colonies was counted with ImageJ software.

## Cellular ROS and and lipid peroxidation detection

Cells were seeded in 12-well plates 12 h prior to treatment, pretreated with 2 μM Fer-1(S7243, Selleckchem, USA), or 32 μg mL⁻¹ CHDV for 24 h. Detection of cellular ROS was based on the peroxide-dependent oxidation of DCFH-DA (BL714A, Biosharp, China) to form a fluorescent compound named dichlorofluorescein (DCF). An hour after irradiation, cells were washed with PBS and incubated with 10 μM DCFH-DA at 37ºC for 30 min. For lipid peroxidation assay, fresh medium containing 5 μM C11-BODIPY dye (D3861, Invitrogen, USA) was added to each well 24 h after irradiation, and the cells were incubated at 37 °C for 30 min. The cells were washed with PBS and trypsinized to obtain a cell suspension. Cellular ROS and lipid peroxidation levels were immediately analyzed by flow cytometry (Canto, BD Biosciences, USA). The visual image of the DCF and C11-BODIPY fluorescence in cells was taken on fluorescence microscopy.

## LDH release assay

LDH cytotoxicity assay kit (C0016, Beyotime, China) was used to verify the ferroptosis-induced cell membrane damage. Cells were seeded into a 96-well plate (5 × 10⁴ cells per well) and treated with CHDV. After 12 h, cells were irradiated (4, 8, 12, and 16 Gy) and allowed to proliferate for 3 more days. Then, 120 μL of each cell culture supernatant was put into a new centrifuge tube and then adding 60 μL of the substrate. After incubation in the dark for 30 min, the reaction was stopped by adding 50 μL of the stop solution. Finally, the LDH release was quantified by monitoring the absorbance at 490 nm.

## Cellular DNA damage detection

One day post irradiation (4, 8, 12, 16 Gy), cells were fixed with 4% paraformaldehyde for 10 min. Then, 0.2% Triton X-100 solution was used to permeate the cell membrane. Next, the blocking reagent was utilized to block the cells for 1 h. Cells were treated with Phospho-Histone H2AX antibody (Ser139, Cell Signaling Technology, USA) for 24 h at 4 °C and treated with fluorescence secondary antibody (Alexa Fluor 488-labeled Goat Anti-Rabbit IgG, Cell Signaling Technology, USA) for another 1 h. Finally, fluorescent images of the cells were captured by fluorescence microscopy.

## Adhesion experiments

FITC-labeled CHDV was used for all adhesion experiments. For in vitro adhesion experiments, mucin solution (30 mg mL⁻¹ in PBS) and transferrin solution (500 μg mL⁻¹ in PBS) were added to 24-well plates and incubated overnight with gentle shaking. After washing, mucin-coated wells (enriched with a negative charges) and transferrin-coated wells (enriched with positive charges) were obtained. CHDV dispersion (1 mg mL⁻¹ in PBS) was added to the wells with gentle shaking for 3 h. After washing the wells three times with PBS, imaging was performed using fluorescence microscopy.

For the ex vivo adhesion tests, animals in both the healthy and radiation colitis groups (n = 4) fasted one day in advance. After euthanasia, 1 cm of colon tissue distal to the anus was dissected. 200 μL of CHDV was suspended in 6 mL of PBS. The colons were placed in the suspension and incubated at 37 °C for 30 min with gentle shaking. After washing three times with PBS, the colons were opened longitudinally with the lumen side facing upwards and finally imaged with an IVIS fluorescence imager (IVIS Lumina Series III, PerkinElmer, USA).

For in vivo adhesion testing, each mouse received 100 μL CHDV by gavage. Animals were euthanized 10 h later, and the whole intestine was dissected without washing. IVIS fluorescence imager was also used for observation.

The collected colons were gently washed once to study the interaction between CDHV and inflammatory gut morphology. The tissue was observed by SEM. The other was fixed, sectioned, and then stained with DAPI Solution (C1005, Beyotime, China) and observed by fluorescence microscopy. The volume fraction of adhered material (Percentage of FITC fluorescence area/total tissue area) was quantified by ImageJ software.

## Animals and ethics statement

C57BL/6 J mice (female, 6–8 weeks old, 18 ± 1 g) were obtained from the Experimental Animal Center of Southern Medical University. Mice were housed in a SPF-grade animal facility with a constant temperature of 24 ± 1 °C, humidity of 51 ± 5%, and a 12-h light/12-h dark cycle. The mice were housed in groups of five per cage and had ad libitum access to food and water, ensuring animal welfare. All animal care and experimental procedures were followed in accordance with guidelines approved by the Ethics Committee of Jinan University Laboratory Animal Center (approval number: 20210528-11). The body weight of the mice did not exceed a decrease of 20% from normal values, meeting the requirements of the Ethics Committee. The mice were anesthetized using pentobarbital sodium and euthanized by overdose anesthesia.

## In vivo radiation colitis model treatment

The mice were randomly grouped (n = 10), including the control group (100 μL of saline), irradiation group (100 μL of saline), irradiation + CHDV group (150 mg kg⁻¹), irradiation + HNTs group (150 mg kg⁻¹), irradiation + CeO$_2$ group (15 mg kg⁻¹), irradiation + DFP group (150 mg kg⁻¹), irradiation + VE group (150 mg kg⁻¹). Except for the control group, all other groups were exposed to 15 Gy irradiation on day 1. This single large dose of irradiation was adopted for inducing radiation colitis effectively. After fasting, the mice were orally administered 8 h before irradiation.

A 5-mm-thick lead plate was used to avoid whole-body exposure, and a window (1 × 1 cm) was opened to allow the colorectal area of the anesthetized mice to be exposed to irradiation. The operating parameters of X-ray irradiation were set as follows: 225 kV/13.33 mA high-energy X-ray, AP-PA technique, dose rate: 1 Gy min⁻¹, a source to specimen shelf distance (SSD): 50 cm, irradiation window: 1 cm² (X-Rad 225 XL, Precision inc., USA). X-rays are emitted from the X-ray source in a cone beam with a beam angle of 40 degrees. The field size at 50 cm SSD is a 28 cm diameter with 90% uniformity over the total area. To achieve uniform dose distribution throughout the animal to a possible depth of >2 cm, a 1-mm Cu filter was used to obtain a filtered (hardened) beam.

Mice were dosed daily for 7 days. During the whole therapeutic period, the body weight of the mice was recorded every day. The mice were euthanatized, and the colons, hearts, livers, spleens, lungs, kidneys, and stomachs were collected at the end of the treatment period. The above tissue was washed with saline and fixed in 4% paraformaldehyde. Animals in the survival analysis groups received the same treatment as described above, and their survival was observed for 30 days to assess survival status.

## Disease activity index score

DAI is the sum of weight loss, diarrhea, and hematochezia scores. The scoring criteria were as follows: no weight loss, loss >1–5%, loss >5–10%, loss >10–15%, loss >15% were scored as 0, 1, 2, 3, and 4 respectively; normal stools, dilute stools, watery stools were scored as 0, 2, and 4, respectively; bloodless stools, occult blood, and severe hematochezia were scored as 0, 2, and 4 respectively.

## MPO activity assay

Colon samples were washed with PBS, frozen and homogenized. The samples were centrifuged at 3000 ×$g$ for 20 min at 4 °C, and the supernatant (5 μL) was added to 200 μL with 100 μL of 0.68 mg mL$^{-1}$ o-anisidine dihydrochloride and 0.1% hydrogen peroxide (both MPO kit components). The absorbance at 460 nm was measured using a spectrophotometer. The results are expressed as MPO activity units per mg of protein.

## MDA assay

Mice colon tissue was collected, homogenized, and lysed. Sample tubes were centrifuged at 10,000–12,000 × $g$ for 10 min at 4 °C, and the supernatant was collected. The protein concentration of each sample was determined by the Pierce™ BCA protein assay kit (23225, Thermo Fisher Scientific, USA). Afterward, samples were assayed for lipid peroxidation levels using the MDA assay kit according to the manufacturer's instructions.

## Q-PCR

Cells (5 × 10$^6$ cells) or colon tissue were lysed by Trizol (15596026, Invitrogen, USA) for 10 min, and supernatants were harvested by centrifugation at 13,200 × $g$ for 12 min. The total RNA was extracted, and cDNA was synthesized with TransScript® Reverse Transcriptase (AT101-02, TransGen Biotech, China). Q-PCR was performed with EasyTaq® PCR SuperMix (AS111-01, TransGen Biotech, China) and ran on CFX Connect Real-Time System (Bio-Rad Laboratories, USA). The primers used for q-PCR were listed in Supplementary Table 2: *Ptgs2*, *Acsl4*, *Fth1*, *Ftl*, *Gpx4*, *Tnf-α*, *Il6*, *Cxcl1*, *Gapdh*.

## Western blot

Cells (5 × 10$^6$ cells) or colon tissue were lysed for 30 min, and lysates were harvested by centrifugation at 13,200 × $g$ for 12 min. Protein concentration was quantified by Pierce™ BCA protein assay kit, and equal amounts of protein were denatured at 100 °C in a loading buffer. Protein samples (~30 μg) were separated on 10% gradient SDS-PAGE gels and transferred to 0.45-μm PVDF membrane (IPVH00010, Millipore, Germany). Membranes were blocked with 5% nonfat milk and then incubated with primary antibodies against the following target proteins: anti-4-HNE (1:1000, ab46545, Abcam, USA), anti-ACSL4 (1:1000, R24265, Zenbio, China), anti-FTH (1:1000, ab65080 Abcam, USA), anti-FTL (1:1000, 10727-1-AP, Proteintech, China), anti-GPX4 (1:1000, ab125066, Abcam, USA), and anti-β-actin (1:3000, ab6276, Abcam, USA). Protein expressions were visualized by incubating the membranes with HRP AffiniPure Goat Anti-Rabbit IgG (H + L) (1:3000, Fude biotech, FDR007) or HRP AffiniPure Goat Anti-Mouse IgG (H + L) (1:3000, Fude biotech, FD0142) and following ECL western blotting substrate (FD8030, Fude Biological Technology, China). The immunoblots were imaged on a Chemiluminescence Image Analysis System (Tanon 5200, Tanon Science and Technology, China).

## In vivo hematological and serum biochemical analysis

Female C57BL/6 J mice (6–8 weeks old) were randomly divided into two groups: the control group (100 μL of saline, i.g.) and the CHDV group (150 mg kg$^{-1}$, i.g.) and administration for a week. Blood samples for hematology analysis were harvested in 10 mL tubes containing sodium EDTA as anticoagulant and hematological parameters were detected by an automatic biochemical analyzer (Sysmex-800, Sysmex

Corporation, Japan). Blood samples for serum biochemistry analysis were collected in 10 mL tubes and then centrifuged at 1000 × $g$ for 10 min to separate the serum. Serum biochemistry was measured by a Beckman Coulter AU680 clinical chemistry analyzer (Beckman Coulter, Miami, USA).

## Histological analysis

The colon tissue (from the cecum to the anus) was harvested and washed away with saline flushing three times. The hearts, livers, spleens, lungs, kidneys, and stomachs for toxicological assessment were also isolated, sectioned 5 μm thick, and stained with hematoxylin and eosin. The photomicrographs were observed by a digital microscope and scanner (M8, Precipoint, Germany).

## Colon iron measurements

Histological analysis was conducted to assess tissue iron accumulation using the Iron Stain Kit (Prussian Blue stain) (ab150674, Abcam, USA), following the manufacturer's instructions. The iron in the tissue reacted with kit components, resulting in a blue color. Images were captured by a digital microscope and scanner (M8, Precipoint, Germany). Total iron was quantified by Iron Assay Kit (ab83366, Abcam, USA). Briefly, the lysed colon homogenate cell was added to 96-well plates for chemical reaction. Iron was released by reducing Fe$^{3+}$ to Fe$^{2+}$ by an acidic buffer. The released iron was then reacted with a chromagen and produced a colorimetric product (at 593 nm) proportional to the total iron level. Total iron concentrations were determined from the standard curve.

## Immunohistochemistry and immunofluorescence

For ACSL4 and 4-HNE immunohistochemistry, 4% paraformaldehyde-fixed paraffin-embedded colon sections were deparaffinized, rehydrated, antigen-retrieved for 20 min in citrate buffer (0.1 M, pH 6) and peroxidase-blocked. Colon sections were incubated with anti-ACSL4 (1:100, R24265, Zenbio, China) and anti-4-HNE (1:100, ab46545, Abcam, USA) diluted in goat serum solution overnight at 4 °C. Next, sections were incubated with horseradish peroxidase-conjugated secondary antibodies and 3,3'-diaminobenzidine. Stained sections were observed by a digital microscope and scanner (M8, Precipoint, Germany).

For immunofluorescence, harvested tissue from the colon were fixed in 4% paraformaldehyde, frozen, embedded, and cryosectioned to obtain 6-μm sections. After rewarming, secondary fixation (4% paraformaldehyde), permeabilization (0.1% Triton X-100/PBS), and blocking with goat serum (CW0130, Cwbio, China), sections were incubated with primary antibodies at 4 °C overnight. Primary antibodies in the blocking buffer were: anti-F4/80 (1:50, ab16911, Abcam, USA), anti-4-HNE (1:50, ab46545, Abcam, USA), and anti-Cleaved Caspase 3 (1:50, 341034, Zenbio, China). Sections were washed three times with PBS to remove non-specific binding. Next, sections were incubated with secondary antibodies at room temperature for 2 h. Secondary antibodies used were: Alexa Fluor 594 donkey anti-rat IgG (H + L) (1:200, A18744, Invitrogen, USA), Alexa Fluor 488 goat anti-rabbit IgG (H + L) (1:200, A11008, Invitrogen, USA), Alexa Fluor 555 goat anti-rabbit IgG (H + L) (1:200, A21428, Invitrogen, USA). After incubation, Sections were then counterstained using DAPI for 10 min and mounted in an antifade mounting medium (P0126, Beyotime, China).

## Statistics and reproducibility

Data were represented as means ± SD. Statistical comparisons of all data were analyzed by GraphPad Prism 8.0 (GraphPad Software, USA), Origin 2021 (OriginLab, USA), FlowJo V10 (BD Biosciences, USA), or ImageJ 1.8.0 (National Institutes of Health, USA). *P* values were determined by unpaired two-tailed *t* tests between two groups or one-way ANOVA with Tukey's post hoc test between multiple groups. Survival was measured via the Kaplan–Meier method, and statistical

significance was calculated by Log-rank test. Values of all significant correlations ($P < 0.05$) were considered statistically significant. For all studies, samples were randomly assigned to various experimental groups. No data were excluded from the analyses. Measurements were taken from distinct samples rather than repeated measuring the same sample. All data were obtained from the results of at least three independent experiments.

## Reporting summary
Further information on research design is available in the Nature Portfolio Reporting Summary linked to this article.

## Data availability
The data that support the findings of this study are available within the main text and its Supplementary Information file. Source data are provided with this paper.

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

## Acknowledgements
The authors thank the National Natural Science Foundation of China (M.L., 52073121, R.H., 82125038), Natural Science Foundation of Guangdong Province (M.L., 2019A1515011509, R.H., 2021B1515120023), Science and Technology Planning Project of Guangzhou (M.L., 202102010117), the Fundamental Research Funds for the Central Universities (M.L., 21622406), the Innovation Team Project of Guangdong Provincial Department of Education (R.H., 2020KCXTD003), Project Team of Foshan National Hi-tech Industrial Development Zone Industrialization Entrepreneurial Teams Program (M.L., 2220197000129), Medical Science and Technology Research Foundation of Guangdong Province (X.L., A2023044), Fellowship of China Postdoctoral Science Foundation (X.L., 2022TQ0122, 2023M731327), and Lift Project of Guangdong Second Provincial General Hospital (X.L., TJGC–2022002).

## Author contributions
M.L., R.H., Y.F., and X.L. conceived and designed the experiment. Y.F., X.L., and Z.L. performed the experiment and analyzed the data. X.F. and Y.W. made substantial contributions to the collection, analysis, and interpretation of clinical data. M.L. and R.H. supervised the project. Y.F. and X.L. wrote the manuscript. All authors discussed the results and commented on the manuscript.

## Competing interests
The authors declare no competing interests.
