## [Peer Review File · Nature Communications]

A Ferroptosis-targeting Ceria Anchored Halloysite as Orally Drug Delivery System for Radiation Colitis TherapyReviewers' Comments:

Reviewer #1:

Remarks to the Author:

[Note from the Editor: please see attached document] 
Reviewer #2:

Remarks to the Author:

The current manuscript by Feng et al titled "A Ferroptosis-targeting Ceria@Halloysite as Orally Drug Delivery System for Radiation Colitis Therapy" is a very interesting study. The authors developed CeO₂@HNTs@DFP nano-formulations and shown that the nanotube alleviated radiation colitis by inhibiting ferroptosis. The authors have shown rigorous experimental design and appropriate data interpretation. Results are convincing, in particular with the use of in vitro and in vivo assays.

Major points:

1. In Figure 1, irradiation could regulate lipid peroxidation and ferroptosis-associated genes in colitis. However, multiple factors could influence the lipid peroxidation or gene expression (Ptgs2, Acsl4, Ftl, and Fth1). Dose ferrostatin-1, a inhibitor of ferroptosis reverse the effects of irradiation?
2. In Figure 5 and Figure 6, the authors also need to demonstrate whether ferrostatin-1 abolished the role of nanotube in irradiation-induced colon injury.
3. The authors showed that ROS was critical for irradiation-mediated ferroptosis. Whether the nanotube may influence irradiation-mediated ferroptosis through the ROS.

Minor points:

1. In Figure 4, the authors used RSL3 to induce ferroptosis in vitro. Why chose the RSL3 in this study? Erastin is also a inducer of ferroptosis.
2. The authors focused on the ferroptosis in intestinal epithelial cells. I know that immune cells (macrophage, ect.) play important role in colitis. If irradiation influence the ferroptosis of macrophages? If the nanotube exerts its effect through regulating the function of macrophages? Please discuss.

Reviewer #3:

Remarks to the Author:

Current manuscript demonstrates the involvement of Ferroptosis in radiation induced colitis. Involvement of ferroptosis in ulcerative colitis is already well known. Various inhibitors for ferroptosis have been observed to have positive effects in attenuating tissue damage associated with colitis. The manuscript does not show any new biological information related to ferroptosis and colitis. This manuscript primarily focused to examine a therapeutic approach which targets three major causes of ferroptosis including iron overload, lipid peroxidation and ROS production. There are multiple concerns in this report including radiation model, insufficient mechanistic study and limited experimental detail resulting moderate enthusiasm about this manuscript.

Major Concerns:

Radiation induced colitis is one of the major chronic problem for pelvic radiotherapy. However, radiation induced intestinal toxicity is not only limited to colitis. Patients undergoing pancreatic SBRT are very often suffers from acute small bowel toxicity which is primarily due to stem cell loss. To understand the involvement of Ferroptosis in radiation induced intestinal toxicity it is important to examine both radiation induced small intestinal and colonic toxicity. Author also did not show which mucosal cell type is more prone to ferroptosis compared to apoptosis. Is there any differences in stem cells vs transit amplifying cells in radiation induced ferroptosis.

In the method section, they have mentioned abdominal irradiation which probably includes small intestine as well unless they irradiated pelvic area only. However, in survival study shown 100% lethality within 15 days of irradiation which primarily results from radiation induced acute small bowel toxicity.

Radiation procedure was not clear. Dose rate, energy level, irradiator model was not described. Exposure field was not described properly.

Clinically relevant Radiation fractionation is critical as single dose will not be used in patients undergoing abdominal radiotherapy.

Detection of ferroptosis related protein was in colonic tissue. However, it is important to determine the cell type undergoing this process. Flowcytometry and/or Microscopic image with Immuno-fluorescence staining will be needed.

Result section describing in vivo therapeutic efficacy is not clear. Again, from experimental detail it is not clear whether mice received whole abdominal irradiation or pelvic irradiation. Next, author mentioned that abdominal irradiation of 15Gy can cause injury to nervous system. Which is not correct. Lethal dose of whole body irradiation can cause damage in nervous system.

Not clear when histopathology was done in figure 5.

No data on inhibition of macrophages recruitment and macrophages producing ROS in response to the therapeutic agent.

No translational relevance of dose range for CHDV.

Several reports shown that radiation induced toxicity primarily mediated through p53/puma mediated apoptosis. Present study failed to show any comparative study of Ferroptosis vs apoptosis.

Hemostatic powders based on inorganic mineral reduces radiation induced colitis. However, present study did not make any attempt to enterocyte specific blocking of iron uptake and thereby examine radiation induced colitis. Moreover, there is no STATISTICAL measurement of iron uptake in enterocyte (FIGURE 6B).

There are mice model available which are prone for iron overload (Hjv^{-/-} and Smad4^{Alb/Alb} mice). it would be more mechanistic and informative to test the drug effect in these mice with exposure to radiation.

What happened with Ferroptosis regulator GPX4 in response to irradiation and therapeutic agent.

Figure 6G number of animals was not mentioned

Figure 5C no p value

Figure 3J no quantification of fluorescence.

Response to the comments of the reviewers:

Reviewer #1:

[Note from the Editor: please see attached document]

Response: We appreciate your careful review and suggestions for this manuscript. The whole text has been revised based on your comments and meticulously checked to avoid any formatting or grammatical deficiencies. We believe the language is acceptable for publication now.

Reviewer #2:

Major points:

1. In Figure 1, irradiation could regulate lipid peroxidation and ferroptosis-associated genes in colitis. However, multiple factors could influence the lipid peroxidation or gene expression (*Ptgs2*, *Acsl4*, *Ftl*, and *Fth1*). Dose ferrostatin-1, an inhibitor of ferroptosis reverse the effects of irradiation?

Response: Thanks for your good suggestion. Ferrostatin-1 (Fer-1), a radical-trapping antioxidant, is a potent and selective inhibitor of ferroptosis *in vitro*. It acts radical-trapping antioxidants to prevent lipid-peroxidation caused membrane damage and thereby inhibits cell death (*Signal Transduction Targeted Ther.*, 2021, 6(1): 49). Accordingly, the protective effect of Fer-1 on radiation-induced ferroptosis was supplemented in the intestinal epithelial cell model. The C11-BODIPY results showed that Fer-1 reversed irradiation-induced lipid peroxidation (Figure S2a, b). It is attributed to effects of Fer-1 scavenging the initiating alkoxy radicals and other rearrangement products (*Proc. Natl. Acad. Sci.*, 2014, 111(47): 16836). As a consequence, irradiation-induced upregulation of ferroptosis-related genes (*Ptgs2*, *Acsl4*, *Ftl*, and *Fth1*) was also reversed by Fer-1 pretreatment (Figure S2c).

However, it was found that Fer-1 have a limited effect on total cellular reactive oxygen species (ROS) (Figure 2d, e), probably because Fer-1 has better affinity for lipid ROS. The results of colony formation also demonstrated the limitation of Fer-1, i.e., although the number of surviving colonies was significantly increased, it was unable to completely reverse the irradiation-induced cell proliferation impairment (Figure 2f, g). In total, although the proof of irradiation-induced ferroptosis is conclusive, using Fer-1 alone cannot completely reverse the irradiation damage. Similar conclusions were drawn from previous studies that although ferroptosis plays a critical role especially in irradiation-resistant cancer cells that are insensitive to DNA damage, the types of irradiation-induced cell death are diverse and include apoptosis, autophagy, and others (*ACS Chem. Biol.*, 2020, 15(2): 469). More synergistic therapeutic systems should be

developed to understand the combined damaging effects of irradiation. The related parts have been modified accordingly.

Figure S2. a) Lipid peroxidation assessment in IEC-6 cells pretreated with DMSO as normal control, 2 μM Fer-1, or 32 $\mu\text{g mL}^{-1}$ CHDV for 24 h followed by exposure to 8 Gy of irradiation. b) Bar graphs showing relative levels of lipid peroxidation by C11-BODIPY staining in the indicated cells, $n = 3$. c) q-PCR analysis of ferroptosis-related genes (*Ptgs2*, *Acs14*, *Ftl*, and *Fth1*) expression in IEC-6 cells pretreated with DMSO (control), 2 μM Fer-1, or 32 $\mu\text{g mL}^{-1}$ CHDV for 24 h followed by exposure to 8 Gy of irradiation, $n = 3$. d) Intracellular total ROS indicated by DCFH-DA of IEC-6 cells pretreated with DMSO as normal control, 2 μM Fer-1, or 32 $\mu\text{g mL}^{-1}$ CHDV for 24 h followed by exposure to 8 Gy of irradiation, and e) the statistics

of mean fluorescence intensity of DCF, $n = 3$. f) Crystal violet staining and g) quantification of the surviving colonies of IEC-6 cells irradiated by 8 Gy X-ray in different treatment groups, $n = 3$. Error bars are means \pm SD. Significant difference was marked as *** $P < 0.001$ vs Control, # $P < 0.05$, ## $P < 0.01$, ### $P < 0.001$ vs X-ray group.

2. In Figure 5 and Figure 6, the authors also need to demonstrate whether ferrostatin-1 abolished the role of nanotube in irradiation-induced colon injury.

Response: Due to the inability to completely reverse radiation damage at the cellular level, as well as the poor plasma stability and imprecise systemic administration of Fer-1, so the *in vivo* studies are debatable. In consideration of the "3R principle" of animal welfare, the cell model was used as a replacement to Fer-1 animal experiments. As shown in Figure S2a, b, it was found that the effect of the nanotube on Fer-1 is superimposed rather than abolished. The nanotube material can scavenge multiple classes of ROS (Figure 2g-i). When combined with Fer-1, synergistic effects can be produced via both the SOD-like and peroxidase-like activities of nanozyme loaded nanotube. The Fenton reaction inhibitory properties of deferiprone and the scavenging of lipid ROS by Fer-1 were confirmed. The expression levels of ferroptosis-related genes were also significantly decreased.

At the level of total cellular ROS (Figure S2d, e), CHDV treatment controlled the mean fluorescence intensity of total cellular ROS probe to a level similar to or even slightly lower than that of healthy controls, indicating effective scavenging of irradiation-induced ROS. Fer-1 is unable to abolish the effects of CHDV due to the incomplete reversal of irradiation effects of Fer-1 described in Question 1. The related parts have been modified accordingly.

3. The authors showed that ROS was critical for irradiation-mediated ferroptosis. Whether the nanotube may influence irradiation-mediated ferroptosis through the ROS.

Response: Thanks for your kind suggestion. Ferroptosis is a death induced by lipid ROS accumulation exceeding a threshold, so the scavenging of ROS is beneficial to alleviate irradiation-induced ferroptosis. To clarify this hypothesis, the ROS and lipid peroxidation changes before and after treatment were supplemented by flow cytometry and the expression of ferroptosis-related genes was detected (Figure S2a-c). These results demonstrate that CHDV can mitigate irradiation-mediated ferroptosis via reducing lipid ROS. These supplementary experiments have been added to Figure S2 and are discussed in the " *In vitro* ferroptosis reversal and evaluation of radiation protection" Section.

Minor points:

1. In Figure 4, the authors used RSL3 to induce ferroptosis *in vitro*. Why chose the

RSL3 in this study? Erastin is also an inducer of ferroptosis.

Response: RSL3 specifically inhibits ferroptosis without bothering other cell death pathways. RSL3 inhibits GPX4 activity by covalent binding, and GPX4 inactivation results in the accumulation of lipid peroxides and eventual ferroptosis (*Cell. Mol. Life Sci.*, 2016, 73(11): 2195). Admittedly, Erastin was also recognized as inhibitor of ferroptosis. Erastin targets the mitochondrial voltage-dependent anion channel protein and inhibits the cystine/glutamate antiporter system X_c^- , thereby inducing ferroptosis. Meanwhile, Erastin can also activate endoplasmic reticulum stress inducing apoptosis (*Cell Death Differ.*, 2016, 23(3): 369). The level of ferroptosis in radiation colitis is positively correlated with the damage severity in this work. The main thrust of this work is to explore the therapeutic effect of ferroptosis inhibition on radiation colitis. Therefore, the specific targeted ferroptosis inhibitor was needed. We agree with you that Erastin will be a good choice in radiation colitis model, especially when the research work develops functional materials that can alleviate both apoptosis and ferroptosis. Reasonable explanation for the use of RSL3 has been supplemented accordingly.

2. The authors focused on the ferroptosis in intestinal epithelial cells. I know that immune cells (macrophage, ect.) play important role in colitis. If irradiation influence the ferroptosis of macrophages? If the nanotube exerts its effect through regulating the function of macrophages? Please discuss.

Response: Irradiation directly induces intestinal epithelial ferroptosis, which is the source of the intense inflammatory response, so this work focuses on the intestinal epithelium. Indeed, macrophages play an important role in radiation colitis, mainly in the later stages of radiation exposure that large numbers of immune cells are induced by inflammatory factors and recruited to the injury site. It is not excluded that irradiation causes ferroptosis in macrophages (*Am. J. Gastroenterol.*, 2000, 95(5): 1221), but since they are mostly recruited at a later stage, only a few macrophages in the colonic mucosa lamina propria undergo irradiation exposure. Thus, the direct damage of irradiation on macrophages was not considered in this study. Considering that after macrophage recruitment, they accumulate excess iron by phagocytosis of erythrocytes at the injury site and aggravate the risk of local ferroptosis, the nanotube material was designed to alleviate macrophage iron stress by relieving bleeding and chelating iron.

According to your suggestion, immunofluorescence co-staining of macrophages (red) and lipid peroxidation product 4-Hydroxynonenal (4-HNE, green) in the radiation colitis (7 days post irradiation exposure) tissues was performed. Significant 4-HNE accumulation and macrophage recruitment was observed in the colitis lesions. The accumulation and site variability of 4-HNE indicated that ferroptosis occurred at the tissue injury sites, and its abundance was positively correlated with the severity of the

injury of villi and tight junctions. These severely injured sites activated inflammation leading to macrophage recruitment and activation. Notably, co-localization of macrophages and 4-HNE fluorescence was rarely observed, suggesting that ferroptosis did not occur in these macrophages. Reduced recruited macrophages and less 4-HNE were observed in the material treatment group, implying that the material inhibited ferroptosis and alleviated inflammation, thereby reducing macrophage recruitment. These supplementary experiments have been added to Figure 1 and are discussed in the "In vivo therapeutic efficacy" Section.

Figure 1e. Immunofluorescence co-staining of macrophages (F4/80, red), lipid peroxidation (4-HNE, green) and DAPI (nuclei, blue) in colon sections.

Reviewer #3:

Current manuscript demonstrates the involvement of Ferroptosis in irradiation induced colitis. Involvement of ferroptosis in ulcerative colitis is already well known. Various inhibitors for ferroptosis have been observed to have positive effects in attenuating tissue damage associated with colitis. The manuscript does not show any new biological information related to ferroptosis and colitis. This manuscript primarily focused to examine a therapeutic approach which targets three major causes of ferroptosis including iron overload, lipid peroxidation and ROS production.

There are multiple concerns in this report including irradiation model, insufficient mechanistic study and limited experimental detail resulting moderate enthusiasm about this manuscript.

Response: We appreciate your thorough understanding of the study of ferroptosis in relation to multiple diseases. As a hot topic nowadays, ferroptosis occurs by different mechanisms in diverse diseases, including ulcerative colitis. Ulcerative colitis is influenced by genetic and environmental factors, while radiation colitis is induced by radiation-induced intestinal barrier damage. Due to different etiologies, ulcerative colitis and radiation colitis cannot be treated in the same way. Therefore, we believe

that understanding radiation colitis from the perspective of ferroptosis is of great significance, and the advances of this work are worthy to be reported.

This work confirms that irradiation dose is positively correlated with ferroptosis levels and mechanistically revolves around lipid peroxidation and iron homeostasis changes. Moreover, it is an interdisciplinary study to treat radiation colitis from the perspective of ferroptosis by medicinal clay and nanozymes. *Nature Communications* is enlightened and inclusive, encouraging the publication of groundbreaking insights in multiple fields such as medicine, cell biology, materials science, mineralogy, and more. We believe this work will inspire researchers' interests in various fields once published in this journal, further studies on the mechanisms and therapeutics of radiation colitis also should be performed.

Major Concerns:

1. Irradiation induced colitis is one of the major chronic problems for pelvic radiotherapy. However, irradiation induced intestinal toxicity is not only limited to colitis. Patients undergoing pancreatic SBRT are very often suffers from acute small bowel toxicity which is primarily due to stem cell loss. To understand the involvement of Ferroptosis in irradiation induced intestinal toxicity it is important to examine both irradiation induced small intestinal and colonic toxicity.

Response: To precisely construct a radiation colitis model, the extent of irradiation exposure was controlled. In detail, a 5 mm thick lead plate was used to avoid whole-body exposure, and a window (1 × 1 cm) was opened to allow the colorectal area of the anesthetized mice to be exposed to irradiation. Following your good suggestions, the H&E staining of the distal small intestine (jejunum) and proximal small intestine (ileum) sections after irradiation was supplemented. Histopathologically, the intestinal villus length and crypt integrity were in the normal range, even in the proximal colon, demonstrating that the small bowel toxicity in this modeling method is controllable. The colonic area showed shortened villi, localized absence of crypts, massive immune cell infiltration, and localized microhemorrhage, exhibiting colitis features. The results have been added in Figure S25 and detailed in the "*In vivo* therapeutic efficacy" Section and "Mice and radiation colitis model treatment " Section.

Figure S25. H&E staining of tissue sections of distal small intestine, proximal small intestine, and colon 7 days after irradiation exposure.

2. Author also did not show which mucosal cell type is more prone to ferroptosis compared to apoptosis.

Response: Thanks for your valuable advice about cell types that undergo apoptosis and ferroptosis after irradiation. The colon sections were stained with intestinal stem cells (ISCs, Lgr 5⁺), ferroptosis (4-HNE), and apoptosis (cleaved-Caspase 3) biomarkers. As shown in the Figure below, Lgr5⁺ intestinal stem cells (ISCs) are located at crypt base. Intestinal cell types are spatially heterogeneous. ISCs reside at the crypt, giving rise to transient amplifying cells that go through continuous steps of proliferation, differentiation, and finally anoikis (a form of programmed cell death) while migrating upwards to the villus tip (*Nat. Rev. Gastroenterol. Hepatol.*, 2018, 15(8): 497).

Immunofluorescence co-staining of intestinal stem cells (Lgr5, red) and DAPI (nuclei, blue) in sections of healthy colon. Red arrows denote intestinal stem cells.

Figure S3. *In vivo* tests for comparative studies of apoptosis/ferroptosis after exposure to irradiation. Immunofluorescence staining for apoptosis (cleaved-Caspase 3) and lipid peroxidation (4-HNE) in colon tissues. Cell nuclei were stained with DAPI to reveal villi and intestinal integrity.

Cleaved-Caspase 3, the end effector of apoptosis (including intrinsic and extrinsic pathways), was chosen to evaluate irradiation-induced apoptosis. As displayed in Figure S3, apoptotic cells were mainly in the epithelial layer of the intestinal mucosa and less frequently in the lamina propria. Few green fluorescence of cleaved-Caspase 3 was found at the crypt base. **These results suggest that irradiation-induced apoptosis is likely to occur in terminally differentiated cells at the end of the villi, but less frequently in the stem cell niche (crypt bottom).** In contrast, ISCs have a greater capacity for DNA damage repair and radio-resistance (*Gastroenterology*, 2012, 143(5): 1266). In addition, p53-mediated apoptosis involves only the stem cell region in the small intestine but is rarely found in the colonic crypt (*Cancer Res.*, 1994, 54(3): 614). The anti-apoptotic gene Bcl-2 is hardly expressed in the small intestine but strongly expressed at the colonic crypt base (*J. Cell Sci.*, 1995, 108(6): 2261). In conclusion, in the irradiated colon, apoptosis occurred predominantly in non-stem cell subsets, and apoptosis sensitivity was enhanced with cell migration toward the end of the villi.

The location of the ferroptosis biomarker 4-HNE in the irradiated colonic mucosa was slightly different from that of cleaved-Caspase 3. They share a low fluorescence intensity at the crypt base but a high intensity at the end of the villi. The difference is that the onset of ferroptosis is not gradual but occurs strongly at the contact surface with the lumen, and at sites where the villi and tight junctions are severely damaged. Combined with the ferritin gene protein changes in the irradiated colon that displayed in Figure 1, ferroptosis may correlate with capillary rupture bleeding at the site of severe

injury. Hemoglobin-induced iron deposition from microvascular rupture exacerbates ROS and lipid peroxidation has been reported in several works (*Aging Dis.*, 2022, 13(5); *Stroke*, 2017, 48(4): 1033). In addition, most of the enterocytes at the top of the villi are absorptive cells. Many studies have shown that nutritional signals regulate ferroptosis (*Int. J. Mol. Sci.*, 2021, 22(22): 12403). For example, dietary iron induces ferroptosis by reducing the expression of SLC7A11 and GPX4 through nuclear factor erythroid 2-related factor 2 (*Front. Oncol.*, 2021, 11: 614778). Ingestion of long-chain polyunsaturated fatty acids and highly fermentable fiber increases the accumulation of cellular lipid ROS, which induces ferroptosis (*Dig. Dis. Sci.*, 2020, 65: 840-851). All these studies imply that enterocytes with high nutrient uptake may be at higher risk of ferroptosis.

These findings confirmed the evidence for the applicability of the nanotube materials. The material was targeted to the mucosal defect by electrostatic interactions and then followed by inhibition of bleeding, iron uptake, and lipid ROS to alleviate ferroptosis, respectively, to treat radiation colitis. These supplementary data and discussions have been added accordingly.

3. Are there any differences in stem cells vs transit amplifying cells in irradiation induced ferroptosis.

Response: Distinguishing stem cells from transit amplifying (TA) cells in radiation-induced ferroptosis is an ambitious topic. Since the research related to intestinal ferroptosis is in its early stage, the association between ISCs, TA cells and ferroptosis has not been reported yet. During the regeneration process of the intestinal epithelium, ISCs constitutively replenish to asymmetrically divide into two daughter cells, one maintaining as a stem cell for self-renewal at the crypt and the other becoming a TA cell to undergo differentiation (*Int. J. Mol. Sci.*, 2020, 22(1): 357). As shown in the Figure below, in terms of crypt ecological niche, Lgr5⁺ crypt base ISCs are intercalated with Paneth cells at the crypt base and continuously generate rapidly proliferating TA cells, which occupy the remainder of the crypt (*Stem Cells Transl. Med.*, 2017, 6(2): 666). Existing theories suggest that TA cells have a faster proliferation and progressive loss of stemness compared to stem cells, while hardly participating in intestinal absorption and secretion. Therefore, studies on the oxidative stress and energy metabolism of TA cells are scarce.

Schematic diagram of intestinal crypt. Small bowel epithelium is organized into villus and crypts. $Lgr5^+$ crypt base columnar cells are intercalated with Paneth cells at the crypt base and continuously generate rapidly proliferating TA cells, which occupy the remainder of the crypt. TA cells differentiate into the various functional cells on the villi (enterocytes, goblet, enteroendocrine, and Paneth cells) (*Stem Cells Transl. Med.*, 2017, 6(2): 666).

Crypt stem cells contain taxa that are hypersensitive to irradiation (they may be in a proliferative state rather than quiescent), a property that functionally protects the stem cell compartment from genetic damage. Damaged stem cells are replaced by the first two to three generations of TA cells, which would have a much better repair capacity, and which would fall back into the stem cell niche at the crypt base while regaining stem cell properties (*Nature*, 1977, 269(5628): 518). Following irradiation exposure, massive enterocyte damage mediates a sudden proliferation of crypt cells, leading to a transient enlargement of the intestinal crypt and subsequent crypt fission to repopulate the intestine (*Cell stem cell*, 2014, 14(2): 149). These theories are consistent with the crypt fission phenomenon observed in this study. Judging from the site of ferroptosis in the intestine, TA cells may not undergo ferroptosis. The underlying reason may be that as progenitor cells, TA cells differ from terminally differentiated cells in lipid metabolism and iron homeostatic signaling. This issue will be explored in the future work.

4. In the method section, they have mentioned abdominal irradiation which probably includes small intestine as well unless they irradiated pelvic area only. However, in survival study shown 100% lethality within 15 days of irradiation which primarily results from irradiation induced acute small bowel toxicity.

Response: We appreciate your kind reminder. Radiographic colitis has well-established modeling methods, which vary slightly depending on the instrumentation conditions at each institution (*Gut*, 2018, 67(1): 97; *Biomaterials*, 2021, 275: 120925). **The modeled mice were exposed to irradiation only in the pelvic area**, while the rest of the body was

shielded by 5 mm lead plates to avoid whole-body exposure. In detail, anesthetized mice were fixed in a supine position on a lead plate, and another lead plate with a window (1 × 1 cm) was tightly fitted to the pelvis of the mice allowing only the colorectal region to be exposed to irradiation through the window. Since the instrumentation we used is geared toward small animals rather than a medical linear gas pedal, it is unable to precisely irradiate the colonic region by computed tomography. Instrumental limitations led to unavoidable bone marrow damage at the pelvis, which was responsible for the mice's death, but the damage to the colon in our method was conclusive. The detailed modeling methods have been supplemented in the Methods section accordingly.

5. Irradiation procedure was not clear. Dose rate, energy level, irradiator model was not described. Exposure field was not described properly.

Response: Thanks for your valuable comments. The operating parameters of X-ray irradiation were set as follows: 225 kV/13.33 mA high-energy X-ray, AP-PA technique, dose rate: 1 Gy min⁻¹, a source to specimen shelf distance (SSD): 50 cm, irradiation window: 1 cm² (X-Rad 225 XL, Precision inc., USA). X-rays are emitted from the x-ray source in a cone beam with a beam angle of 40 degrees. The field size at 50 cm SSD is a 28 cm diameter with 90% uniformity over the total area. To achieve uniform dose distribution throughout the animal to a possible depth of > 2 cm, a 1 mm Cu filter was used to obtain a filtered (hardened) beam. These parameter details have been supplemented to the Methods section accordingly.

6. Clinically relevant irradiation fractionation is critical as single dose will not be used in patients undergoing abdominal radiotherapy.

Response: Many thanks for your insightful comment. We acknowledge that irradiation fractionation is crucial in clinical because it allows time for normal cells to repair themselves between treatments, thereby reducing side effects. In addition, the significance of irradiation fractionation is aimed at effectively killing cancer cells. Since a group of malignant cells are at various points in their cell cycle, delivering the entire dose of irradiation in a single fraction is ineffective against a proportion of the tumor cells. Dividing the total dose of irradiation into multiple fractions maximizes the probability of irradiating cells when they are in the most radiosensitive period of their cell cycle (*Br. J. Radiol.*, 1971, 44(521): 325). Oxygen-deprived cancer cells are less susceptible to the effects of irradiation. The interval between irradiation treatments allows relatively hypoxic cells to improve their oxygen supply and thus become more sensitive to the subsequent irradiation dose (*Cancer Metastasis Rev.*, 2007, 26: 241).

In contrast, animal models of irradiation-induced tissue damage often use a single large dose of irradiation to create disease model. Romesser et al. created a irradiation-induced gastrointestinal syndrome model by administering a single dose of 14.5 Gy

irradiation to the whole abdomen of mice (*Proc. Natl. Acad. Sci.*, 2019, 116(41): 20672). Han et al. performed 13 Gy whole-body irradiation to produce irradiation-induced multiorgan damage (*Adv. Mater.*, 2020, 32(31): 2001566). Jee et al. performed single doses of 40, 50, and 60 Gy irradiation to construct irradiation proctitis models with different damage degrees (*Biomaterials*, 2021, 275: 120925). In summary, although irradiation fractionation is necessary for clinical radiotherapy, single doses are more widely used in animal irradiation injury models. Supplementary explanations have been added accordingly.

7. Detection of ferroptosis related protein was in colonic tissue. However, it is important to determine the cell type undergoing this process. Flowcytometry and/or Microscopic image with Immuno-fluorescence staining will be needed.

Response: We strongly agree with you that it is important to confirm the type of cells that undergo ferroptosis. Looking at the anatomical regions occurring in ferroptosis as discussed in the immunofluorescence images in Question 2, ferroptosis occurs extensively in the mucosa and is more intense at the tip of the colonic crypt. There was no significant 4-HNE accumulation fluorescence around the niche of Lgr5⁺ ISCs at the base of the crypt. Immunofluorescence images of macrophages and 4-HNE showed no significant co-localization, implying that recruited macrophages are not the major taxa of ferroptosis.

Taken together, the terminally differentiated intestinal absorptive cells are considered to be the main cell type undergoing ferroptosis. The amounts are the most and the site of iron uptake (*Food Funct.*, 2014, 5(7): 1320). The uptake and translocation of iron leads to high iron loading and thus a greater susceptibility to ferroptosis in response to radiation-induced oxidative stress. The relationship between ferroptosis and enterocytes will be further explored subsequently. The relevant discussion has been supplemented to the "Ferroptosis mechanism of radiation colitis" section.

8. Result section describing in vivo therapeutic efficacy is not clear. Again, from experimental detail it is not clear whether mice received whole abdominal irradiation or pelvic irradiation. Next, author mentioned that abdominal irradiation of 15 Gy can cause injury to nervous system. Which is not correct. Lethal dose of whole-body irradiation can cause damage in nervous system.

Response: The treatment effects have been described in more detail accordingly. Detailed modeling methods for mice receiving pelvic irradiation rather than whole body exposure have also been added to the Methods Section (see Question 3). The discussion of neurological damage caused by 15 Gy irradiation is a result of other publications and not a conclusion of this work. We apologize for this mistake and have deleted the corresponding inappropriate discussion.

9. Not clear when histopathology was done in figure 5.

Response: The sampling times (7 days treatment) of tissues have been supplemented in the Experimental section and indicated them on the animal experiment schematic in Figure 5.

10. No data on inhibition of macrophages recruitment and macrophages producing ROS in response to the therapeutic agent.

Response: Following your invaluable suggestion, the macrophage-related results have been supplemented. The effect of therapeutic agents on macrophage recruitment and response to ROS was analyzed in sections of the post-treatment colon by immunofluorescence co-staining of macrophages (F4/80), lipid peroxidation (4-HNE) and DAPI (nuclei). As illustrated in the Figure S27, macrophages resident in the normal colon are located primarily in the subepithelial mucosal lamina propria and lymph nodes or outside the muscularis, maintaining immune homeostasis of the mucosa (*Am. J. Physiol. Gastrointest. Liver Physiol.*, 2016, 311(1): G59). When exposure to irradiation causes colitis, macrophages are recruited to regions of active mucosal inflammation. They are involved in the development and regression of colitis, particularly in the clearance of dead enterocytes, red blood cells produced by microvascular rupture, and invasive intestinal microorganisms. After therapeutic agent treatment, recruited macrophages were significantly reduced due to the decrease of inflammation sites. Some macrophages remained in crypt microvessels, possibly due to overfermented gut microbes in the colitis microenvironment.

In addition, lipid peroxidation levels were significantly increased in the radiated colon, especially at mucosal defects, which is consistent with previous theories stating that ferroptosis is complemented with inflammation levels. Macrophages pool at sites of lipid peroxidation but barely overlap with the green fluorescence of 4-HNE, suggesting that macrophages are recruited to scavenge iron-dead cells without undergoing ferroptosis themselves. This recruitment signal has been documented in our previous study that an oxidized phospholipid on the plasma membran manipulates the eat-me signal and navigates phagocytosis by targeting TLR2 on macrophages (*Cell Death Differ.*, 2021, 28(6): 1971). As the therapeutic agent scavenges ROS and prevents ferroptosis, inflammation and cell death are alleviated after treatment. This results in a more intact colonic mucosa with less lipid peroxidation and reduced macrophage recruitment signals.

Figure S27. Immunofluorescence co-staining of macrophages (F4/80, red), lipid peroxidation (4-HNE, green) and DAPI (nuclei, blue) in colony sections after irradiation with or without therapeutic agent treatment.

Since the increase in ROS in macrophages *in vivo* is transient and difficult to capture, high intracellular ROS in RAW 264.7 macrophages cell line was induced to identify whether the therapeutic agent inhibits ROS producing by macrophages. ROS (e.g., H₂O₂) at the injury site can drive the differentiation of monocytes to macrophages and induce high intracellular ROS in macrophages. It is a notion that ROS in macrophage is essential for uptake and clearance of dying cells; however, maintaining high level of ROS may be detrimental to macrophage or even cause cell death (*Oxid. Med. Cell. Longevity*, 2016). Therefore, H₂O₂ was used to induce macrophages undergoing high oxidative stress to evaluate the effect of the therapeutic agent CHDV on intracellular ROS in macrophages. The flow cytometry results showed that CHDV treatment significantly reduced ROS levels in H₂O₂-stimulated RAW 264.7 cells, indicating that the therapeutic agent was also effective in liberating the high oxidation state of macrophages (Figure shown as below). These supplementary experiments have been added to Figure S27 and are discussed in the “*In vivo* therapeutic efficacy” Section.

Intracellular ROS of RAW 264.7 macrophage line was measured by flow cytometry (DCFH-DA). H₂O₂ (5 μM) was used to stimulate macrophages.

11. No translational relevance of dose range for CHDV.?

Response: Thanks for your valuable advice. Animal models of radiation colitis allow highly controlled experimental conditions, detailed organ insights, standardized, clinically-relevant treatment, and the invention of new therapies. Up to now, few results from radiation colitis animal models have been translated into clinical management and treatments. Because of the differences in manifestations of patients with radiation colitis, such as radiotherapy dose, adjuvant chemotherapy, underlying disease, and patient characteristics (e.g., age, genetics, body composition), the translational relevance of the dose range is challenging.

According to your recommendations, the recommended translational dose range for CHDV is considered effective from 10 to 30 mg/kg (adults) a day. The calculations were derived from the translation of mouse doses to humans and took into account the therapeutically effective doses of the main components of CHDV. For example, antidiarrheal clay is safe and effective within 128 mg/kg, and the dosage should be adjusted according to the severity of diarrhea. Deferiprone at an oral dose of 75 mg/kg/day is safely used to treat iron overload in patients with thalassemia major, while in healthy individual, it is used as a ferroptosis inhibitor in reduced doses to avoid iron deficiency (*Mult. Scler. J.*, 2007, 13(9): 1118).

Moreover, nanozymes with low toxicity, high efficiency, and reasonable administration are crucial to bridging the gap between the rapid development and successful clinical translation of nanozymes (*ACS Appl. Nano Mater.*, 2020, 3(2): 1043). Although it was controversial, ceria was considered a low-toxicity or biocompatible material in previous reports. The reduced particle size and less agglomeration of ceria synthesized *in situ* at the clay nanotube interface enhanced ROS scavenging performance. Besides, oral administration fully considers the clinical situation of radiation colitis to maximize the therapeutic effect and minimize side effects.

Overall, translating innovative therapeutic modalities to safe clinical applications

in humans remains challenging, a problem common to all fields of basic research and nanomedicine. This work explores a mechanistically innovative and effective approach for treating radiation colitis, which is a pioneering and inspiring attempt at the interdisciplinary intersection of biomedicine, materials science, and mineralogy. We believe the inspirational value of this work is endorsed by *Nature Communications* and will attract the continued interest of researchers in multiple fields and advance the clinical treatment of radiation colitis. The discussions of the translational relevance have been supplemented accordingly.

12. Several reports shown that irradiation induced toxicity primarily mediated through p53/puma mediated apoptosis. Present study failed to show any comparative study of Ferroptosis vs apoptosis.

Response: Based on your comments, the comparative study of ferroptosis vs apoptosis have been supplemented. As in the response to Question 2, immunofluorescence staining pictures of irradiated colon sections show apoptosis occurring over a large area of the intestinal epithelium, more in the villi than in the crypt. And ferroptosis also occurred extensively, but more intensely at the severely damaged mucosa and at the tip of the colonic villi exposed to chyme.

Subsequently, by using the ferroptosis inhibitor Fer-1 and the apoptosis inhibitor Z-VAD, the reversal of radiation damage by inhibition of ferroptosis/apoptosis was evaluated. Total cell death levels evaluated by MTT (Figure S3a) showed that Fer-1 rescued cell viability from irradiation damage of $59 \pm 3\%$ to $74 \pm 5\%$, and Z-VAD was slightly higher, rescuing up to $77 \pm 5\%$. The results of membrane damage evaluated by LDH (which occurs more often in iron-dead cells) showed that Fer-1 treatment significantly reduced the LDH release rate to $53 \pm 3\%$ compared to the $70 \pm 4\%$ caused by irradiation, which provided good protection against radiation membrane damage. Interestingly, Z-VAD also reduced LDH release to $57 \pm 3\%$, and the combination of Fer-2 and Z-VAD more dramatically avoided plasma membrane damage (Figure S3b). These results show that irradiation-induced apoptosis is more dominant, but ferroptosis is also widespread, and by inhibiting both apoptosis and ferroptosis, the protection against irradiation-induced cell death can be superimposed.

We strongly agree with you that irradiation induces apoptosis is well established. Irradiation energy directly damages DNA, and cells are repaired or proceed to apoptosis under the regulation of the p53-mediated DNA repair signaling pathway. Besides, ionized water molecules produce large amounts of ROS and directly damaging cells through local peroxidation events. Since irradiation-induced apoptosis has been thoroughly studied, it is more relevant to consider radiation colitis from a new perspective in this work.

Moreover, although the mechanisms of irradiation damage are well known, clinical

radioprotection measures are often ineffective, and new treatment modalities are worth investigating. The efficacy of apoptosis-inhibiting agents such as 5-androstenediol and the free radical trapping agent amphotericin is unsatisfactory. Radioprotective agents alone also fail to alleviate patient complications such as inflammation, diarrhea, and intestinal bleeding in specific diseases such as radiation colitis. Therefore, the discovery of ferroptosis in radiation colitis in this work may develop an alternative therapeutic pathway to compensate for the existing treatment. Using nanozyme-clay as an antidiarrheal agent that simultaneously traps while scavenging ROS may alleviate complications along with radioprotection. The comparative studies of apoptosis vs ferroptosis has been supplemented in Figure S3 and discussed accordingly in the “Introduction” and “Ferroptosis mechanism of radiation colitis” Section.

Figure S3 b c. *In vitro* tests for comparative studies of apoptosis / ferroptosis after exposure to irradiation. b) Cell viability and c) LDH release of IEC-6 cells pretreated with DMSO as normal control, 2 μ M Fer-1, 25 μ M Z-VAD, or combination at 48 h after exposure to irradiation (16 Gy). The data are shown as mean \pm SD, $n = 6$. The data were analyzed by one-way ANOVA with Tukey’s post hoc test. Significant difference was marked as **** $P < 0.0001$.

13. Hemostatic powders based on inorganic mineral reduces irradiation induced colitis. However, present study did not make any attempt to enterocyte specific blocking of iron uptake and thereby examine irradiation induced colitis. Moreover, there is no STATISTICAL measurement of iron uptake in enterocyte (FIGURE 6B).

Response: Thanks for your valuable advice. Admittedly, blocking iron absorption can effectively reduce the iron content in tissues, thus avoiding ferroptosis triggered by iron overload. This work discovered increased gene expression and protein levels of ferritin in colonic tissue after radiation (Figure 1b-d), suggesting altered iron transport in the colon. Moreover, as the response in Question 2, accumulation of 4-HNE was observed at the tip of the colonic villi where mainly absorptive cells. These changes may result from excess iron produced by the oxidation of hemoglobin (leaking from intestinal bleeding) by excess superoxide. Oxyhemoglobin reacts with irradiation-induced H_2O_2 , generates a ferrylhemoglobin $[Hb(4^+)=O]$ species, and upon the introduction of a second H_2O_2 , produces methemoglobin and superoxide. Excess superoxide may destroy the porphyrin and release the iron causing tissue iron overload (*Front. Cell.*

Neurosci., 2020, 14: 603043).

Gene knockout or silencing methods have not been used to block cellular iron uptake because blocking intestinal iron uptake can be achieved directly and safely by oral administration of deferiprone, which has been included in the therapeutic agent. DFP can chelate free iron when outside the cell membrane; it can also cross the membrane and chelate the intracellular free "redox-active" iron pool (*Cell. Mol. Life Sci.*, 2016, 73: 2405), regardless of the enterocyte specific blocking of iron uptake.

Iron deposits indicated with Perls Prussian Blue decreased after CHDV treatment (Figure 6b). In addition, CHDV treatment significantly decreases the gene and protein levels of ferritin (Figure 6d-g), which regulating iron metabolism, suggesting that excessive iron absorption and storage in the intestine is blocked.

According to your suggestion, the total iron content in irradiated colon tissue was added using an Iron Assay Kit to support Figure 6. As illustrated in Figure 6c, it was discovered that a significant increase in the total iron content in colon tissues after irradiation, while CHDV treatment effectively reversed this phenomenon. This result and discussion have been added to the "Ferroptosis as a target for protection against radiation colitis" Section.

Figure 6c. Histogram of colonic iron quantification analysis. The data are shown as mean \pm SD ($n = 3$). The data were analyzed by one-way ANOVA with Tukey's post hoc test. Significant difference was marked as *** $P < 0.001$ vs Control, ## $P < 0.01$ vs X-ray group.

14. There are mice model available which are prone for iron overload ($Hjv^{-/-}$ and $Smad4^{Alb/Alb}$ mice). it would be more mechanistic and informative to test the drug effect in these mice with exposure to irradiation.

Response: Admittedly, absorptive enterocytes and macrophages supply most of the serum iron primarily by recycling metals from senescent erythrocytes, and hemochromatosis models can elevate their ferroportin levels, resulting in increased intestinal iron absorption and elevated serum iron levels (*J. Clin. Invest.*, 2005, 115(8): 2187). We agreed with you that $Hjv^{-/-}$ and $Smad4^{Alb/Alb}$ mice models would be more mechanistic and informative to investigate the therapeutic effect of CHDV.

However, both *Hjv*^{-/-} and *Smad4*^{Alb/Alb} mice develop severe systemic iron overload due to impaired hepcidin expression in the liver (*Hepatology*, 2017, 66(2): 449). Chronically increased oxidative stress from iron overload in the body may increase radiation sensitivity by decreasing cellular oxygen radical scavenging capability (*Radiat. Res.*, 2000, 153(6): 844). Therefore, the systemic damage caused by the radiation superimposed hemochromatosis is tentatively unknown, which creates uncertainty in the prognosis of oral gastrointestinal administration of CHDV. The related part has been modified accordingly.

15. What happened with Ferroptosis regulator GPX4 in response to irradiation and therapeutic agent.

Response: Indeed, the lipid hydroperoxidase GPX4 is a crucial ferroptosis regulator that protects cells against membrane lipid peroxidation and maintains redox homeostasis. According to your suggestions, the expression of the GPX4 gene and protein in response to irradiation and therapeutic agent CHDV was examined. As illustrated in Figure 6d-g, it was observed that the gene and protein levels of GPX4 exhibited a significant decrease after radiation exposure, implying that radiation induces a surge of phospholipid hydroperoxides in membranes and lipoproteins in the colon. After treatment with CHDV, GPX4 levels were significantly restored, indicating that the ROS scavenging and antioxidant function of CHDV effectively reduced lipid peroxidation in the irradiated colon. The results and discussion of these updates have been added to Figure 6d-g and the “Ferroptosis as a target for protection against radiation colitis” Section.

Figure 6d-g. d) The gene expressions of ferroptosis pivotal genes such as *Ptgs2*, *Acsf4*, *Ftl*, *Fth1*, and *Gpx4* in colon tissue ($n = 3$), and e) they were quantified and visualized by heatmap. f, g) The protein expressions of ACSL4, FTL, FTH, 4-HNE, and GPX4 in colon tissue were determined by western blotting analysis, $n = 3$. The data are represented by mean \pm SD. The data were analyzed by one-way ANOVA with Tukey’s post hoc test. Significant difference was

marked as ** $P < 0.01$, *** $P < 0.001$ vs Control, # $P < 0.05$, ## $P < 0.01$, ### $P < 0.001$ vs X-ray group.

16. Figure 6G number of animals was not mentioned

Response: The number of animals in Figure 6G have been added and the whole text have been checked to avoid similar mistakes.

17. Figure 5C no p value

Response: The p-values in Figure 5C have been added.

18. Figure 3J no quantification of fluorescence.

Response: The quantitative statistical charts of Figure 3J have been supplemented.

Reviewers' Comments:

Reviewer #2:

Remarks to the Author:

I appreciate the authors for the manuscript improvement.

Reviewer #3:

Remarks to the Author:

Authors have addressed all the comments in the revised version. However, there are couple of points still needs to be addressed considering the clinical/translational importance of the study.

1. In this study authors examined the prophylactic role of CHVD. Will it be also successful if animals receiving CHVD treatment post irradiation. Mitigation of radiation induced colitis is very crucial for any future clinical application.

2. Author have established that inhibition of ferroptosis could be a major therapeutic strategy against radiation induced colitis. It is true that majority of preclinical studies in radiation biology are adopting large dose single fraction. Unlike radiation accident/explosion and countermeasure development any studies related to cancer treatment side effects demands fractionated radiation. There are enough evidences that biology of fractionated irradiation is different from single fraction large dose. Therefore, ferroptosis as etiology of radiation induced colitis may not be true in clinical scenario. Which further dampen the relevance of this study.

Response to the comments of the reviewers:

1. In this study authors examined the prophylactic role of CHVD. Will it be also successful if animals receiving CHVD treatment post irradiation. Mitigation of radiation induced colitis is very crucial for any future clinical application.

Response: We agree with the reviewer that mitigation of radiotherapy-induced proctitis (therapeutic effect) is important for clinical applications. Indeed, pre-administration post irradiation is needed for preventive effect evaluation and continuous administration after irradiation is appropriate for therapeutic and mitigation effect evaluation. However, whether the use of antioxidants to mitigate irradiation damage is clinically prophylactic or therapeutic has not been clearly defined. If irradiation can be predicted, pre-exposure administration may be superior to post-exposure administration. The drug onset time will differ depending on the administration method and individual differences. Considering acute radiation injury often occurs several minutes to hours after exposure, patients usually take medicine upfront to minimize damage and continued after exposure.

Administration of CHDV by the mice in this work is not just prophylaxis. The duration of therapeutic dosing was much longer than the prophylactic dosing. Only 1 dose was administered before irradiation and administered daily after irradiation for 7 days. Pre-administration 8 hours prior to irradiation is in consideration of the relatively longer onset time required for oral agents, and for predictable radiotherapy, aggressive pre-administration provides better mitigation of injury. Additionally, a single dose has limited effect, so continuous administration after irradiation is considered to be the critical for CHDV. Due to non-native English writing, "prevent" is synonymous with "avoid" and "protect" in the Chinese context, and we apologize for any misinterpretation caused by the difference in terms. Based on your comment, the use of "prevent" has been reduced, and animal experimental procedures has described in more details accordingly in the “*In vivo therapeutic efficacy*” Section.

2. Author have established that inhibition of ferroptosis could be a major therapeutic strategy against radiation induced colitis. It is true that majority of preclinical studies in radiation biology are adopting large dose single fraction. Unlike radiation accident/explosion and countermeasure development any studies related to cancer treatment side effects demands fractionated radiation. There are enough evidences that biology of fractionated irradiation is different from single fraction large dose. Therefore, ferroptosis as etiology of radiation induced colitis may not be true in clinical scenario. Which further dampen the relevance of this study.

Response: We greatly appreciate your comments about the biological differences

between preclinical studies and clinical reality of fractionated irradiation. Indeed, fractionation utilized the difference in repair efficiency between normal and tumor cells in fractionation, which enhanced tumor cell casualties and reduced normal cell damage. As a result, fractionated irradiation is less frequently used in radiation damage studies of normal tissues due to the possible unsuccessful modeling. The biological differences between large dose single fraction and fractionated radiation are mainly in the severity rather than existence or not.

Coincidentally, our ongoing fundamental and clinical studies have provided clinical evidence to address your questions. Colonic sections from clinical patients suffering from radiation colitis caused by fractionated radiotherapy were used for ferroptosis evaluation. Sections from 5 patients were evaluated, and colon irradiated sites were compared with non-irradiated sites in the same patients. As shown in the Figure below, immunohistochemical results for ACSL4 and 4-HNE displayed that ferroptosis occurred extensively in all patients' irradiated sites. As an end product of ferroptosis, 4-HNE abundantly accumulates around the villi (patient 1) and also in sites of the severely deficient intestinal mucosa (patient 4), indicating that cells in these sites have undergone ferroptosis. As an upstream lipid metabolism-regulating enzyme of ferroptosis signaling, ACSL4 was more extensively expressed after irradiation exposure, containing crypt cells. This phenomenon suggests that irradiation induces widespread lipid peroxidation. Both of these ferroptosis biomarkers increase with the severity of the lesion sites. The results of these clinical samples demonstrate that fractionated radiotherapy also induces lipid peroxidation leading to ferroptosis, and is highly correlated with the severity of radiation colitis.

Taken together, these data have provided evidence to support that ferroptosis is an etiologic factor in radiation colitis, whether fractionated irradiation or single fraction large dose irradiation. We maintain the conclusion that ferroptosis was involved in radiation colitis, and the ferroptosis-related therapies are promising. Admittedly, fractionated radiotherapy is more commonly employed than the single high-dose radiotherapy used in this study to allow for the repair of normal tissues. However, it does not significantly alleviate the occurrence of ferroptosis. Both precise radiotherapy and ferroptosis-suppressive therapies promise to minimize normal tissue damage. We believe this work will be illuminating and of broad interest to ferroptosis research in the field of radiation colitis.

The Methods section has been updated to include the ethical considerations, data collection, and source of clinical data for this retrospective study. The study protocol followed ethical guidelines and was approved by the Human Medical Ethics Committee of the Sixth Affiliated Hospital of Sun Yat-sen University. Clinical data were collected

from 5 patients diagnosed with radiation colitis, who underwent radiotherapy between January 2021 and August 2021. The data included in the study were obtained from the hospital's records and were de-identified to ensure patient confidentiality. The retrospective nature of the study allowed for the use of existing data and the requirement for obtaining informed consent was exempted by the Ethics Committee. The patients involved in the study were not part of any clinical trials, and the clinical data were collected as part of standard care practices.

The representative clinical data and the relevant discussion have been supplemented accordingly in Figure 1, "*Ferroptosis mechanism of radiation colitis*" Section, and "*Ferroptosis as a target for protection against radiation colitis*" Section. The complete immunohistochemistry results have been added to the Supplementary Information. The two Figures below present updated clinical data.

Figure 1. Ferroptosis occurs in radiation colitis. a) Schematic showing the ferroptosis-related pathways under radiation colitis. b) Representative immunohistochemical staining of 4-HNE and ACSL4 of colonic tissue samples from patients with radiation colitis, and their normalized expression intensity (relative to non-irradiation), $n = 5$. c) Western blotting analysis of ferroptosis-related protein expression in colon tissue of mice. d) Normalized protein expression (relative to β -actin) levels. e) The ferroptosis-related gene expressions of *Ptgs2*, *Acsl4*, *Ftl*, and *Fth1* in colon tissue. f) Immunofluorescence images (F4/80, 4-HNE, DAPI) of colon tissue. g) Cell viability ($n = 3$), and h) lactate dehydrogenase (LDH) release after incremental dose irradiation exposure ($n = 3$). i) Fluorescence images of reduced (red) and oxidized (green) C11-BODIPY, j) Representative flow cytometry histogram of oxidized C11-BODIPY, and k) their mean fluorescence intensity quantification ($n = 3$). Error bars are means \pm standard deviation (SD). * $P < 0.05$, ** $P < 0.01$, *** $P < 0.001$ vs the 0 Gy or non-irradiation group.

Supplementary Figure

Figure S1. 4-HNE and ACSL4 are upregulated in human colon tissue from patients with radiation colitis. a, b) Representative immunohistochemical staining of 4-HNE and ACSL4 in colonic tissue samples from patients with radiation colitis. Scale bars: 200 μ m.

Reviewers' Comments:

Reviewer #3:

Remarks to the Author:

Authors have responded to all the comments.

Addition of human samples significantly improved the importance of this paper.
Please provide the post irradiation time point of these samples.

Response to the comments of the reviewers:

For Reviewer #3:

1. Please provide the post irradiation time point of these samples.

Response: Thank you for your kind feedback. We have provided information about the post irradiation time point of these clinical samples. Additionally, we have included more clinical features of the patients, including the times of fractionated radiotherapy and symptoms etc. Please refer to Table S3 for the details. The discussion has been supplemented accordingly in “*Ferroptosis mechanism of radiation colitis*” Section.

Supplementary Table 3. Clinical features of 5 patients with radiation colitis

Number	Primary carcinoma	Fractionated radiotherapy, Times	Clinical diagnosis	Symptom	Duration between the first symptom to the last radiotherapy, Months	Duration between the last radiotherapy and radiation colitis surgery, Months
1 (Supplementary Fig. 1)	Endometrial cancer	28	Radiation colitis	Hematochezia	2	5
2 (Supplementary Fig. 1)	Endometrial cancer	6	Radiation colitis	Increased stool frequency, hematochezia, anal pain	4	7
3 (Supplementary Fig. 1)	Cervical cancer	25	Radiation colitis	Increased defecation frequency, hematochezia,	0, 6	13
4 (Supplementary Fig. 1)	Cervical cancer (IIA)	25	Radiation colitis	Increased defecation frequency, dark red bloody stools	1	4
5 (Figure 2b)	Cervical cancer (IB)	25	Radiation colitis	Abdominal pain, anal pain, increased frequency of defecation, vaginal defecation	8	9